# An artificial cell capable of signal transduction mediated by ADRB2 for the regulation of glycogenolysis

Yanhao Liu[1,2,3], Wan Zhao[1,2,3], Yingming Zhao[1,2,3], Xiangxiang Zhang[1,2,3], Jingjing Zhao[1,2,3], Shubin Li[1,2,3] ✉, Yongshuo Ren[1,2,3] & Xiaojun Han ●[1,2,3] ✉

Bottom-up construction of artificial cells helps elucidate the working mechanism of cells. Signal transduction from extracellular to intracellular artificial cells is essential for autonomous artificial cells. It remains highly challenging to reconstitute G protein-coupled receptor (GPCR) signaling pathways to regulate downstream metabolism in artificial cells. Here, we reconstitute β2-adrenergic receptor, Gs subunit α and adenylate cyclase V into artificial cell membranes to enable signal transduction from extracellular isoproterenol (ISO) to intracellular cAMP (visualization via Epac1-cAMP probes). cAMP production is ISO dose-dependent, with a maximum amplification fold of $22.45 \pm 2.14$. By encapsulating the glycogenolytic pathway, cAMP activates protein kinase A, triggering phosphorylation of phosphorylase kinase and glycogen phosphorylase to convert glycogen to glucose-1-phosphate (G-1-P). G-1-P is further converted to 6-phosphogluconolactone accompanying with NADPH. ISO stimulation induces G-1-P and NADPH generation, achieving progressive signal amplification. The successful reconstitution of GPCR-mediated signaling pathway in artificial cells paves the way for developing autonomous artificial cells.

Artificial cells are cell-like structures capable of mimicking partial (or whole) functions of living cells[1–3]. The construction of artificial cells from scratch helps elucidate the working mechanism of cells[4,5]. The ultimate goal of artificial cell construction is the creation of an autonomous cell-like structure that can communicate with its environment for metabolism, growth and self-reproduction[6–8]. Signal transduction from extracellular to intracellular artificial cells is the core function of autonomous artificial cells.

Previous attempts at the signal transduction of artificial cells have relied mainly on synthetic receptors, which mimic the working mechanism of natural receptors, including receptor tyrosine kinases (RTKs) and G-protein-coupled receptors (GPCRs)[9–11]. RTKs transduce signals through ligand-induced receptor dimerization mechanisms.

Cysteine-modified cholesterol[12] and lithocholic acid derivatives[13] were synthesized as receptors to mimic RTKs, which were dimerized via supramolecular interactions to transduce signals from outside artificial cells to inside them. Anionic ethylenediamine-based receptors were reconstituted into artificial cell membranes, which were dimerized by extracellular cationic ligands via electrostatic interactions to generate intracellular fluorescent signals[14]. Transmembrane DNA strands were also used as RTK analogues anchored on the vesicle membrane[15]. Extracellular DNA-induced dimerization triggered intracellular DNAzyme generation, cleaving quenched substrates to produce fluorescence. The conformational changes of GPCRs were mimicked with synthesized azobenzene-modified foldamers containing photoresponsive properties for signal transduction in artificial

[1]State Key Laboratory of Urban-rural Water Resource and Environment, School of Chemistry and Chemical Engineering, Harbin Institute of Technology, Harbin, China. [2]MIIT Key Laboratory of Critical Materials Technology for New Energy Conversion and Storage, School of Chemistry and Chemical Engineering, Harbin Institute of Technology, Harbin, China. [3]Heilongjiang Provincial Joint Laboratory of Molecular Science (International Cooperation), School of Chemistry and Chemical Engineering, Harbin Institute of Technology, Harbin, China. ✉e-mail: lishubin@hit.edu.cn; hanxiaojun@hit.edu.cn

cells[16]. Extracellular light induced local conformational changes in the photosensitive head group of the foldamer, inducing global conformational chiral changes in the receptors. Cholesterol-modified triplex DNA (TD), as a synthetic receptor, transmits pH signals across lipid bilayers via H+-mediated TD conformational transitions[17]. These receptors fail to generate secondary messengers to regulate biological downstream metabolism.

GPCRs constitute the largest family of membrane receptors that mediate responses to a wide variety of hormones, neurotransmitters, and other environmental stimuli[18–20]. GPCRs, including the β2-adrenergic receptor (ADRB2)[21–23], A$_{2A}$ adenosine receptor (A$_{2A}$AR)[24,25], and dopamine receptor[26], were reconstituted into phospholipid vesicle membranes to study their conformational changes stimulated by ligands and ligand-binding properties. Upon agonist binding, transmembrane helix VI of A$_{2A}$AR exhibited rotation and outwards displacement in the membrane of lipid vesicles[25]. Dopamine receptor D2 was reconstituted into the membrane of polymer vesicles with the half maximal effective concentration (EC$_{50}$) of dopamine to be 30 μM[27]. No secondary messengers were produced inside artificial cells. The regulation of intracellular downstream metabolism through GPCR signal transduction triggered by extracellular ligands remains a great challenge in the field of artificial cells.

Herein, we demonstrate artificial cells capable of signal transduction mediated by β2-adrenergic receptors are constructed by coreconstituting the β2-adrenergic receptor-Gs subunit α (ADRB2-Gsα) complex with adenylate cyclase V (ADCY5), which transduces the extracellular primary messenger (isoproterenol, ISO) to intracellular second messengers (adenosine cyclophosphate, cAMP). cAMP subsequently regulates the downstream glycogenolytic pathway inside artificial cells to obtain glucose-1-phosphate (G-1-P). A natural complex cellular signaling pathway is precisely replicated in artificial cells, which paves the way for autonomous artificial cells.

## Results

The concept of this paper is schematically illustrated in Fig. 1. Three membrane proteins, including a GPCR protein of β2-adrenergic receptor (ADRB2), Gs subunit α (Gsα), and adenylate cyclase V (ADCY5), are reconstituted into artificial cell membranes to obtain artificial cells capable of signal transduction mediated by ADRB2. Upon extracellular stimulation of ADRB2 by ISO (the primary messenger), Gsα is activated with the replacement of GDP by GTP, which subsequently activates ADCY5 to produce the secondary messenger cAMP (Fig. 1a, Eq. (1) in 1b). Thus, the signal is transduced into intracellular artificial cells by the ADRB2-Gsα complex and ADCY5. cAMP subsequently regulates the downstream glycogenolytic pathway, which involves protein kinase A (PKA), phosphorylase kinase (PhK), glycogen phosphorylase (PYGM), phosphoglucomutase (PGM) and glucose-6-phosphate dehydrogenase (G6PDH) (Fig. 1a). After cAMP binds to PKA, it triggers a phosphorylation cascade involving PhK and PYGM. Glycogen is catalyzed by PYGM to break down into glucose-1-phosphate (G-1-P) (Eq. (2) in Fig. 1b). The conversion of G-1-P to glucose-6-phosphate (G-6-P) is catalyzed by phosphoglucomutase (PGM). G-6-P

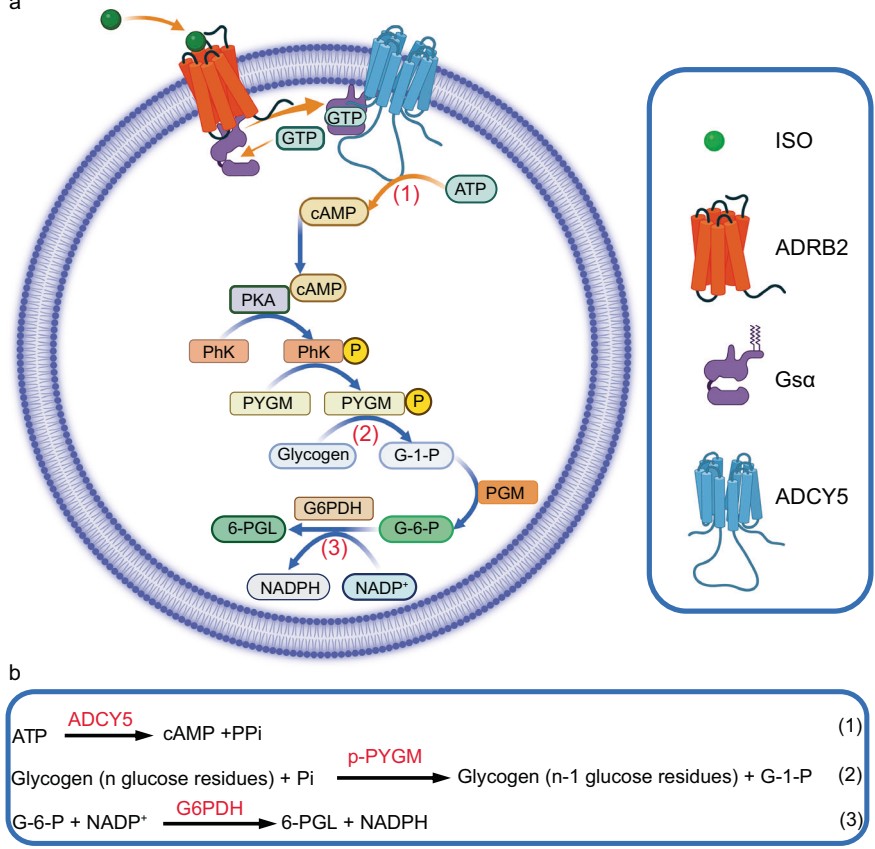

**Fig. 1 | Schematic illustration of an artificial cell capable of signal transduction mediated by ADRB2 for the regulation of glycogenolytic metabolism. a** The signal transduction from extracellular ISO into intracellular cAMP and subsequent regulation of glycogenolytic metabolic pathway inside an artificial cell; ISO: isoproterenol; ADRB2: β2-adrenergic receptor; Gsα: Gs subunit α; ADCY5: adenylate cyclase V; PKA: Protein kinase A; PhK: Phosphorylase kinase; PYGM: Glycogen phosphorylase; PGM: Phosphoglucomutase; G6PDH: Glucose-6-phosphate dehydrogenase; GDP: Guanosine diphosphate; GTP: Guanosine triphosphate; ATP: Adenosine triphosphate; cAMP: Cyclic adenosine monophosphate; G-1-P: Glucose 1-phosphate; G-6-P: Glucose-6-phosphate; 6-PGL: 6-Phosphoglucono-lactone; NADP+(H): Nicotinamide adenine dinucleotide phosphate. Created in BioRender. Liu, Y. (2025) https://BioRender.com/689u8j4. **b** Chemical reaction equations involved in the signal transduction and glycogenolytic metabolism; ADCY5: Adenylate cyclase V; p-PYGM: Phosphorylated glycogen phosphorylase.

is subsequently catalyzed by glucose-6-phosphate dehydrogenase (G6PDH) to produce 6-phosphogluconolactone (6-PGL), accompanied by the reduction of NADP$^+$ to NADPH in the artificial cells (Eq. (3) in Fig. 1b).

### Signal transduction pathway from ISO to cAMP in the solution

To elucidate the signal transduction pathway mediated by ADRB2, plasmids containing ADRB2 (Supplementary Fig. 1), Gsα (Supplementary Fig. 2) and ADCY5 (Supplementary Fig. 3) were designed and transferred into Sf9 cells for overexpression. The molecular weights of ADRB2, Gsα, and ADCY5 matched their theoretical molecular weights of 47 kDa[28] (Fig. 2b), 45 kDa[29] (Fig. 2c), and 102 kDa[30] (Fig. 2d), respectively, which implied the successful purification of those proteins. The ADRB2-Gsα complex was obtained by collecting the first fraction after eluting the mixture of ISO-bound ADRB2 and GDP-bound Gsα with size exclusion chromatography (Fig. 2e) for signal transduction.

The signal transduction pathway from ISO to cAMP mediated by ADRB2 was established in solution by mixing the ADRB2-Gsα complex and ADCY5 (Fig. 2a). The stimulation of ISO to ADRB2-Gsα caused the binding of GTP to Gsα, consequently activating ADCY5 to convert ATP to cAMP. By measuring cAMP production, ADCY5 exhibited the greatest catalytic activity (Supplementary Fig. 4) when 10.0 μM ISO and 2.0 mM Mg$^{2+}$ were added at pH 8 and 30 °C (Fig. 2f). 37 °C was chosen as the reaction temperature, since the catalytic activity of ADCY5 at 37 °C was not significantly different from that at 40 °C (Fig. 2g). At a fixed pH of 8, a temperature of 37 °C and an ISO concentration of 10.0 μM, the optimum Mg$^{2+}$ concentration was 2.0 mM (Fig. 2h). Therefore, the conditions of pH 8, 37 °C and 2.0 mM Mg$^{2+}$ were chosen for the following experiments. Under these conditions, the Michaelis constant and maximal velocity values of ADCY5 catalysis of ATP to cAMP were determined to be 144.5 μM and 171.4 μM·min$^{-1}$, respectively (Fig. 2i), which were better than the reported values (620.0 μM and 12.8 μM·min$^{-1}$, respectively)[31]. The molar ratio of the ADRB2-Gsα complex and ADCY5 significantly influenced the production of cAMP (Fig. 2j). A molar ratio of 17.4:10.0 resulted in a maximum cAMP concentration of 133.38 ± 2.09 μM. With a pH of 8, a temperature of 37 °C, a Mg$^{2+}$ concentration of 2.0, and an ADRB2-Gsα complex/ADCY5 molar ratio of 17.4:10.0, the influence of the primary messenger (ISO) concentration on the production of secondary messengers (cAMP) was systemically investigated from $10^{-5}$ to $10^3$ μM (Fig. 2k). cAMP production gradually increased with time and stabilized after 25 min at all the ISO concentrations (Fig. 2k). The cAMP concentration at 25 min gradually increased from $10^{-4}$ μM to 1.0 μM and stabilized above 1.0 μM (Fig. 2l). With the stimulation of 1.0 μM ISO, cAMP production reached 128.30 ± 2.62 μM, with an ATP conversion yield of 25.66%. Signal transduction from ISO to cAMP involving the ADRB2-Gsα complex and ADCY5 was successful in solution.

### Reconstitution of ADRB2, Gsα and ADCY5 on giant unilamellar vesicles

To construct artificial cells capable of ADRB2 signal transduction, the reconstitution of ADRB2, ADRB2-Gsα, and ADCY5 on giant unilamellar vesicles (GUVs) was essential. To visualize the reconstituted ADRB2 on the GUV membrane, it was covalently tagged with red fluorescent Cy5-maleimide. The red GUV image (Fig. 3a, left image) confirmed the successful reconstitution of ADRB2 on the GUV membrane, while the GUV was labeled with green fluorescent NBD-PE (Fig. 3a, middle image). The merged image (Fig. 3a, right image) indicated that both Cy5-maleimide-labeled ADRB2 (Cy5-ADRB2) and NBD-PE were located in the membrane. The percentage of ADRB2-GUVs was determined by flow cytometry by varying the molar ratio of Cy5-ADRB2 and lipids (Supplementary Fig. 5). The percentage of ADRB2-GUV increased in a concentration-dependent manner to 96.29 ± 2.25% with an increase in the ADRB2:lipids molar ratio from 1:$10^6$ to $10^3$:$10^6$ (Fig. 3b). Beyond a

molar ratio of $10^3$:$10^6$, no significant improvement was observed. Therefore, $10^3$:$10^6$ was chosen as the optimal ratio for ADRB2 reconstitution. The reconstituted ADRB2 in the GUV membrane was in a fluid state according to fluorescence recovery after photobleaching (FRAP) measurements (Supplementary Fig. 6). The diffusion coefficient of ADRB2 in the GUV membrane was determined to be 7.81 × $10^{-9}$ cm$^2$/s, which was higher than that of ADRB2 on the natural cell membrane ($10^{-10}$–$10^{-11}$ cm$^2$/s). Compared with natural cell membranes, artificial cell membranes lack the constraints of the cytoskeletal network[32] and lipid rafts[33] because of their simple lipid compositions (DOPC/DOPG/CHS), which explains the faster diffusion coefficient of ADRB2 in GUV membranes.

The correct orientation of ADRB2 in GUV membranes (the extracellular domain facing outwards) is critical for signal transduction. The extracellular domain of ADRB2 contains an ISO-binding domain and other domains modified with N-glycans. To determine the correct orientation rate of ADRB2 in GUV membranes, we treated GUVs with the addition of PNGase F to cleave glycosidic bonds between asparagine residues and the N-acetylglucosamine of ADRB2. Subsequently, the GUVs were lysed with Triton X-100 to measure the molecular weight of ADRB2. The decrease in molecular weight confirmed the correct orientation of ADRB2 in the GUV membranes (Fig. 3c, top row images), with a correct orientation rate of 94.06 ± 4.24% determined by measuring the band intensities (Fig. 3c, bottom graph). Quantitative analysis following established protocols[34] revealed an average number of receptors of 1.8 × $10^6$ ADRB2 per GUV. The high correct-orientation efficiency and high abundance of ADRB2 in the GUV membrane lay a solid foundation for subsequent signal transduction from extracellular to intracellular artificial cell areas.

After the optimal reconstitution molar ratio of ADRB2 to lipids was determined, ADRB2-Gsα complexes were reconstituted in GUV membranes. BODIPY TR GTPγS was employed as a fluorescent probe to validate ADRB2-Gsα complex reconstitution in the GUV membrane (Fig. 3d). In the absence of binding to Gsα, BODIPY TR-GTPγS aggregated inside the lumen of artificial cells; therefore, they did not emit fluorescence because of the quenching effect (Fig. 3d, left image). Upon ISO stimulation, Gsα was activated to recruit BODIPY TR GTPγS, which consequently emitted red fluorescence (Fig. 3d, right image). With the addition of 1.0 μM ISO, red fluorescence corresponding to BODIPY TR-GTPγS was observed in the GUV membrane (Fig. 3e), confirming the successful reconstitution of the ADRB2-Gsα complex on the GUV. The fluorescence intensity of BODIPY TR-GTPγS increased gradually from 0.5 to 5 min and stabilized after 5 min (Fig. 3f, red curve). With no ISO stimulation, no red fluorescence was observed in the GUV membrane (Fig. 3f, blue curve). These findings validated the successful reconstitution of the ADRB2-Gsα complex in the GUV membrane and, more importantly, the changes in the conformation of Gsα induced by ISO stimulation.

The secondary messenger cAMP is generated from ATP, catalyzed by ADCY5. ADCY5 was reconstituted on GUV to produce cAMP. To visualize the reconstituted ADCY5 in the GUV membrane, it was labeled with green fluorescent FITC. The green fluorescence of the GUV image (Fig. 3g, left image) confirmed the successful reconstitution of ADCY5, while the GUV was labeled with red fluorescent TR-DHPE (Fig. 3g, middle image). Colocalization of both the green fluorescence of FITC-labeled ADCY5 (FITC-ADCY5) and the red fluorescence of TR-DHPE was observed in the GUV membrane (Fig. 3g, right image). The percentage of ADCY5-GUVs was determined by flow cytometry with varying molar ratios of FITC-ADCY5 and lipids (Fig. 3h). ADCY5-GUV populations increased as the molar ratio increased from $10^{-1}$:$10^6$ to $10^3$:$10^6$. The percentage of ADCY5-GUV was 97.53 ± 1.79% at a $10^3$:$10^6$ molar ratio (Fig. 3i). The reconstituted ADCY5 in the GUV membrane was also in a fluid state according to the FRAP measurement (Supplementary Fig. 7). The diffusion coefficient of ADCY5 on artificial cell membranes was 8.92 × $10^{-9}$ cm$^2$/s. The high mobility of

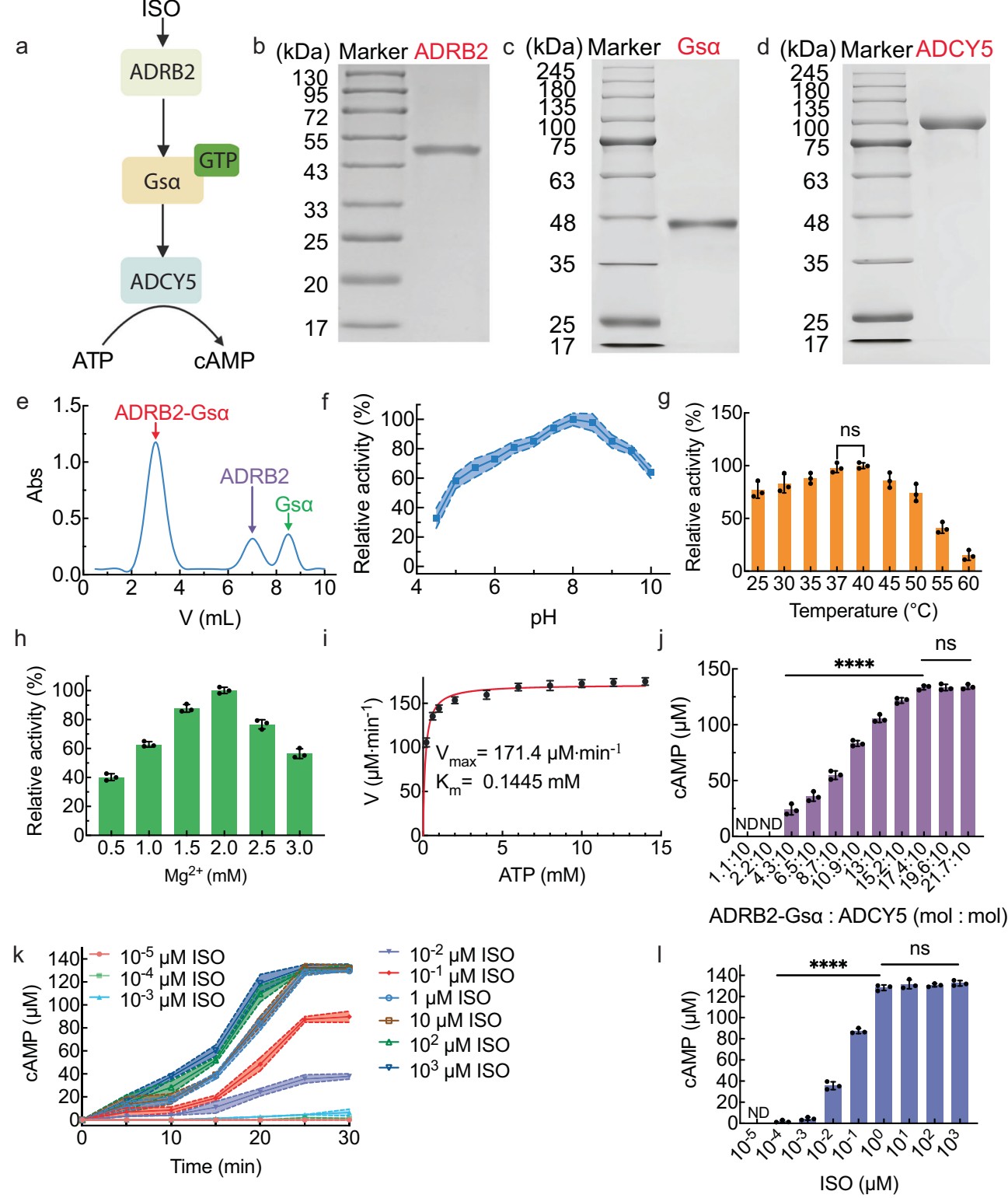

ADRB2 and ADCY5 in artificial cell membranes facilitated their interactions for signal transmembrane transduction.

## Artificial cells capable of signal transduction from extracellular ISO to intracellular cAMP

ADRB2-Gsα complexes and ADCY5 were ready to be reconstituted into artificial cell membranes to construct artificial cells capable of transducing extracellular ISO signals to the intracellular second messenger cAMP (Fig. 4a). No penetration of ISO across the lipid bilayer

membrane was detected (Supplementary Fig. 8h). The coreconstitution of ADRB2-Gsα complexes (Fig. 4b, left image) and ADCY5 (Fig. 4b, middle image) on artificial cells was visualized by fluorescence microscopy. Flow cytometric analysis demonstrated that 97.4% of the artificial cells were reconstituted with both ADRB2-Gsα complexes and ADCY5 (Supplementary Fig. 9), which further confirmed the successful reconstitution of these two proteins.

To monitor intracellular cAMP production in artificial cells, Epac1-cAMP Fret probes were purified. The molecular weight of Epac1-cAMP

**Fig. 2 | Signal transduction pathway from ISO to cAMP mediated by ADRB2 in the solution. a** Scheme of the signal transduction pathway from ISO to cAMP mediated by ADRB2. SDS-PAGE images of purified ADRB2 (**b**), Gsα (**c**) and ADCY5 (**d**) from *Homo Sapiens*. $n = 3$ independent replicates. **e** The size exclusion chromatography of ADRB2, Gsα, and ADRB2-Gsα complex. **f** The effect of pH on the ADCY5 activity by measuring the production of cAMP at 30 °C, 10.0 μM ISO and 2.0 mM $Mg^{2+}$. **g** The effect of temperature on the ADCY5 activity by measuring the production of cAMP at pH 8, 10.0 μM ISO and 2.0 mM $Mg^{2+}$. The data were expressed as mean ± SD, $n = 3$ independent replicates. **h** The $Mg^{2+}$ concentration effect on the ADCY5 activity by measuring the production of cAMP at 37 °C, pH 8 and 10.0 μM ISO. The data were expressed as mean ± SD, $n = 3$ independent replicates. **i** Initial reaction velocities of cAMP production catalyzed by ADCY5 as a function of ATP concentration from 0 to 14.0 mM in the solution (10.0 μM ISO,

2.0 mM $Mg^{2+}$, 6.0 μg/mL ADRB2-Gsα complex, 6.0 μg/mL ADCY5, pH 8) at 37 °C. The data were expressed as mean ± SD, $n = 3$ independent replicates. **j** The cAMP production as a function of ADRB2-Gsα/ADCY5 molar ratios with ADCY5 fixed at 10 nM and ADRB2-Gsα varied from 1.1 to 21.7 nM in a solution (10.0 μM ISO, 2.0 mM $Mg^{2+}$, pH 8) at 37 °C. The data were expressed as mean ± SD, $n = 3$ independent replicates. ****$P < 0.0001$, ns$P > 0.05$, two-sided, multiple comparisons by One-way ANOVA test. **k** ISO dose-dependent cAMP production as a function of time in a solution (17.4 nM ADRB2-Gsα, 10.0 nM ADCY5, $10^{-5}$–$10^3$ μM ISO, 2.0 mM $Mg^{2+}$, pH 8) at 37 °C. **l** The corresponding cAMP production at 25 min of (**k**). The data were expressed as mean ± SD, $n = 3$ independent replicates. ****$P < 0.0001$, ns$P > 0.05$, two-sided, multiple comparisons by One-way ANOVA test. $P < 0.05$ was considered statistically significant. Source data are provided as a Source Data file.

matched the theoretical molecular weight of 153 kDa (Supplementary Fig. 10). The Epac1-cAMP Fret probes interacted with cAMP to change their conformation to enlarge the distance between enhanced yellow fluorescent protein (EYFP) linked to N-terminal and enhanced cyan fluorescent protein (ECFP) linked to C-terminal, leading to the simultaneous intensity increase of ECFP and the intensity decrease of EYFP (Supplementary Fig. 11). At an artificial cell density of $10^7$ cells·$mL^{-1}$, the addition of 1.0 μM ISO to the extracellular solution induced cAMP production within the artificial cells, leading to gradual recovery of ECFP fluorescence (Fig. 4c, top row images) and concurrent attenuation of EYFP emission (Fig. 4c, bottom row images). The FRET ratio of ECFP over EYFP increased gradually (Fig. 4d), which indicated continuous cAMP production. The FRET ratios of Epac1-cAMP from fluorescence spectroscopy were used to determine the cAMP concentration inside the artificial cells (Supplementary Fig. 12). With the addition of 1.0 μM ISO, the FRET ratio of Epac1-cAMP gradually increased from 0 to 20 min and plateaued after 20 min at a concentration of 24.03 ± 1.01 μM (Fig. 4e, blue curve, and Supplementary Fig. 13). In the negative control group without the ADRB2-Gsα complex, no cAMP production was observed from 0 to 30 min (Fig. 4e, orange curve). After 30 min, the cAMP concentration inside artificial cells ($10^7$ cells·$mL^{-1}$) exhibited a three-phase positive correlation with the concentration of ISO from 0.1 to 6.0 μM (Fig. 4f, green curve, and Supplementary Fig. 14). At subthreshold agonist levels (0.1–0.4 μM ISO), almost no cAMP production was observed, indicating insufficient ADRB2 stimulation by ISO. In the activation phase (0.4–6.0 μM ISO), cAMP production gradually increased to a maximum cAMP concentration of 57.42 ± 2.64 μM. Beyond 6.0 μM ISO (saturation phase), no additional cAMP accumulation occurred despite increasing ISO concentrations, demonstrating complete saturation of ADRB2 activation states. This triphasic dose–response profile aligns with classic GPCR signaling dynamics. In the absence of the ADRB2-Gsα complex, no cAMP was generated after stimulation with different concentrations of ISO for 30 min (Fig. 4f, orange curve). These results indicate that extracellular ISO stimulation did not induce intracellular cAMP production in the absence of the ADRB2-Gsα complex. The generation of cAMP was triggered entirely by the specific binding of ISO to ADRB2. The signal amplification peak occurred at 1.0 μM ISO (22.45 ± 2.14-fold) (Fig. 4g), since the slope at that point was the greatest.

Signal transduction from extracellular ISO to intracellular cAMP is regulated by an antagonist (alprenolol) and an inverse agonist (carazolol) that specifically bind to ADRB2. Alprenolol competed with ISO to inhibit signal transduction. After 1.0 μM ISO was added to the artificial cell solution for 10 min, 1.0 μM alprenolol was added to replace the bound ISO with ADRB2, causing no cAMP accumulation inside artificial cells from 10 to 20 min (Fig. 4h, i). With the addition of a high concentration of ISO (10.0 μM) at 20 min, cAMP was produced again because of the reoccupation of ISO with ADRB2 (Fig. 4h, i). The inverse agonist carazolol deactivated ADRB2 to stop signal transduction. Similarly, 1.0 μM inverse agonist carazolol was added to the artificial solution at 10 min to permanently inhibit the production of cAMP.

Even the addition of 10.0 μM ISO did not restore the accumulation of cAMP (Fig. 4j, k). The results from fluorescence spectroscopy indicated the exact same pattern (Fig. 4l) as that from the fluorescence microscopy. No significant difference in the final cAMP concentration was observed between artificial cells with a diameter of 3 μm and those with a diameter of 10 μm (Supplementary Figs. 15–17). From the abovementioned results, it can be concluded that artificial cells capable of transducing extracellular primary messenger (ISO) to intracellular secondary messenger (cAMP) mediated by ADRB2 were successfully established. Thus, the regulation of downstream metabolic pathways inside artificial cells was further studied.

## Construction of glycogenolytic pathway

cAMP is a central secondary messenger and coordinates diverse physiological metabolic pathways, including the glycogenolytic pathway. Glycogenolysis is a vital metabolic process in organisms that efficiently and rapidly provides energy for life activities. To regulate the glycogenolytic pathway inside artificial cells through the established ISO signal transduction pathway, a glycogenolytic pathway in solution was constructed. The glycogenolytic pathway involves 5 enzymes that convert glycogen to NADPH and 6-PGL (Fig. 5a). cAMP activated PKA to phosphorylate PhK to p-PhK, which subsequently phosphorylated PYGM to p-PYGM. Glycogen was split into G-1-P by p-PYGM. The conversion of G-1-P into G-6-P was catalyzed by PGM. G-6-P was subsequently converted into 6-PGL, and NADPH was produced.

PhK phosphorylation was confirmed by western blot images of p-PhK recognized by its specific antibody (Fig. 5b). The phosphorylation percentage of PhK was quantified by measuring the intensities of the p-PhK band over the PhK+p-PhK band. It increased with increasing PKA enzymatic units from 0 to 1.0 U at a fixed PhK of 1.0 U to reach a value of 93.67 ± 2.52% and subsequently plateaued (Fig. 5c). Thus, 1.0 U of PKA and 1.0 U of PhK were chosen for the phosphorylation of PYGM to obtain p-PYGM, which was confirmed by western blot images of p-PYGM recognized by its specific antibody (Fig. 5d). The phosphorylation percentage of PYGM increased as the number of PhK enzymatic units increased from 0 to 1.4 U at a fixed PYGM concentration of 1.0 U to reach a value of 76.30 ± 2.86%, and this value remained constant thereafter (Fig. 5e). Thus, 1.0 U of PKA, 1.4 U of PhK and 1.0 U of PYGM were chosen for glycogenolysis. The phosphorylation of PYGM was also influenced by the ATP concentration (Fig. 5f). The phosphorylation percentage of PYGM gradually increased as the ATP concentration increased from 0 to 0.8 mM, but decreased above 0.8 mM (Fig. 5g). At an ATP concentration of 0.8 mM, the phosphorylation percentage was 90.84 ± 2.58%. The decline of phosphorylation percentage at higher concentrations of ATP may be caused by the conformation change of PhK[35,36] to prevent its further phosphorylation, consequently decreasing the production of p-PYGM. Thus, 0.8 mM ATP, 1.0 U of PKA, 1.4 U of PhK, and 1.0 U of PYGM were selected for subsequent glycogenolysis.

With the addition of 1.0 μM cAMP and 200.0 μM $NADP^+$, NADPH production gradually increased from 0 to 8 min, with a yield of

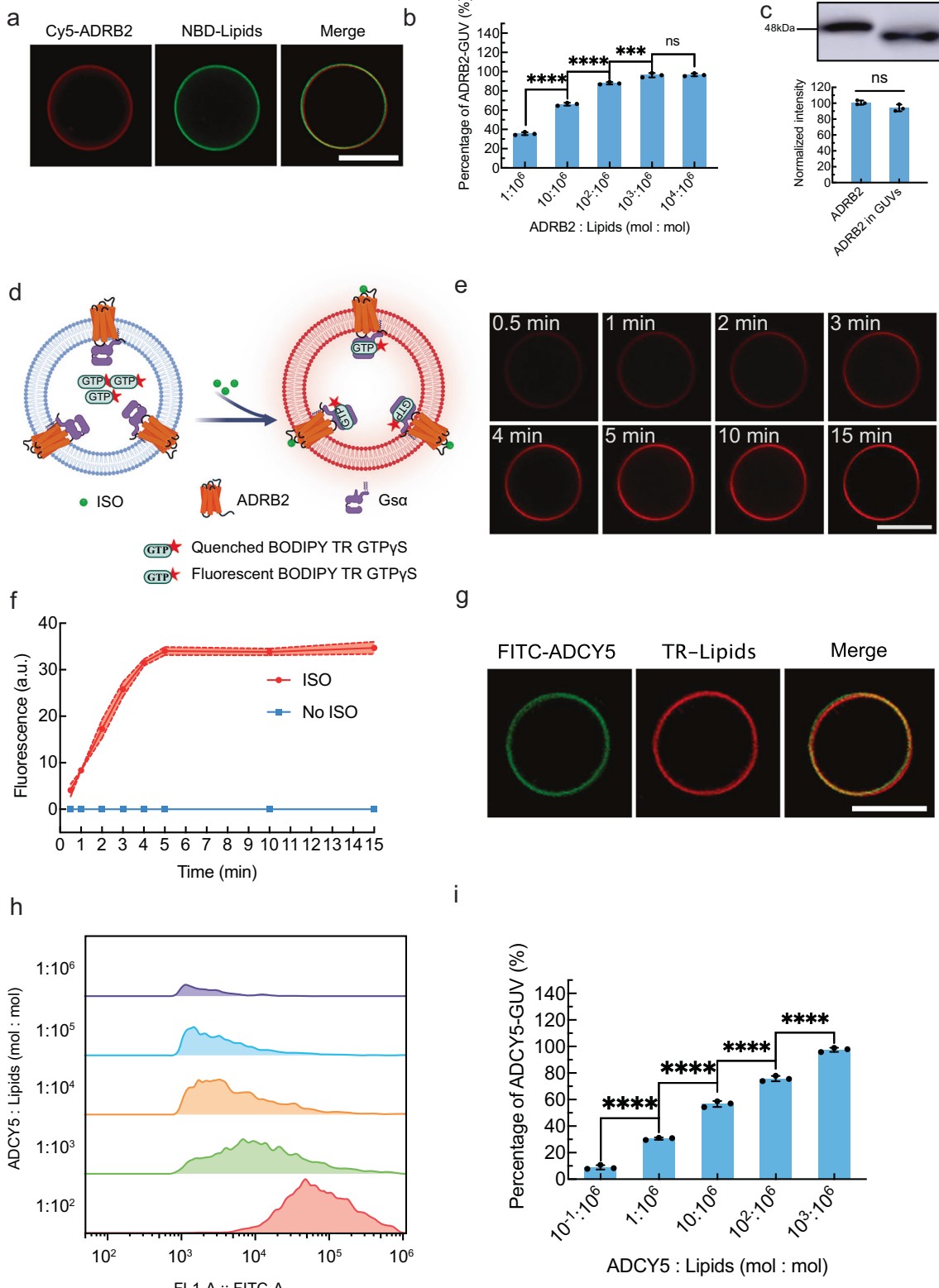

178.32 ± 8.18 μM NADPH (Fig. 5h), which indicated the successful construction of the glycogenolytic pathway in solution.

The glycogenolytic pathway was encapsulated in artificial cells. Artificial cells capable of converting glycogen to G-1-P were constructed with a lumen solution composed of 1.0 μM cAMP, 0.8 mM ATP, 0.5 mM glycogen, 1.0 U/mL PKA, 1.4 U/mL PhK, and 1.0 U/mL PYGM. The reaction was initiated by rapidly increasing the

temperature from 4 °C to 37 °C. The concentration of G-1-P gradually increased from 0 to 8 min and plateaued at a maximum value of 63.28 ± 3.07 μM (Supplementary Fig. 18, purple curve). No G-1-P was produced in the absence of PKA (Supplementary Fig. 18, orange curve) or PhK (Supplementary Fig. 18, blue curve). Artificial cells capable of converting glycogen to 6-PGL were constructed by further encapsulating 0.2 mM NADP+, 1.0 U/mL PGM, and 1.0 U/mL G6PDH. In the

**Fig. 3 | The reconstitution of ADRB2, Gsα and ADCY5 on GUVs. a** The representative fluorescence images of Cy5-ADRB2-GUV (with NBD-PE in the bilayer) taken with red channel (left image), green channel (middle image), and their merged image (right image) at ADRB2 and lipids molar ratio of $10^3$:$10^6$. The scale bar is 10.0 µm. **b** The percentage of ADRB2-GUVs with varied ADRB2 to lipids molar ratios ranging from 1:$10^6$ to $10^4$:$10^6$. The data were expressed as mean ± SD, $n = 3$ independent replicates. ****$P < 0.0001$ 1:$10^6$ vs. 10:$10^6$, ****$P < 0.0001$ 10:$10^6$ vs. $10^2$:$10^6$, ***$P = 0.0007$ $10^2$:$10^6$ vs. $10^3$:$10^6$, $^{ns}P > 0.05$ $10^3$:$10^6$ vs. $10^4$:$10^6$, two-sided, multiple comparisons by One-way ANOVA test. **c** The percentage of ADRB2 with correct orientation on GUVs determined by WB (top image) and band intensities (bottom graph). The data were expressed as mean ± SD, $n = 3$ independent replicates. **d** Schematic diagram of the Gsα detection mechanism using BODIPY TR GTP$_\gamma$S. Created in BioRender. Liu, Y. (2025) https://BioRender.com/g8zgjzy. **e** The representative time-series images of GUVs stimulated by 1.0 µM ISO for the binding events of BODIPY TR GTP$_\gamma$S to Gsα from 30 s to 15 min. The scale bar is 10.0 µm.

$n = 3$ independent replicates. **f** The corresponding fluorescence intensity of GUV in (**e**) (red curve) and the fluorescence intensity of GUV with no ISO stimulation (blue curve). $n = 3$ independent replicates. **g** The representative fluorescence images of FITC labeled ADCY5-GUV (with TR-DHPE in the bilayer) taken with green channel (left image), red channel (middle image), and their merged image (right image) at ADCY5 and lipids molar ratio of $10^3$:$10^6$. The scale bar is 10.0 µm. $n = 3$ independent replicates. **h** The flow cytometry histogram of FITC-ADCY5-GUVs with varied ADCY5 to lipids molar ratios ranging from $10^{-1}$:$10^6$ to $10^3$:$10^6$. $n = 3$ independent replicates. **i** The percentage of ADCY5-GUVs with varied ADCY5 to lipids molar ratios ranging from $10^{-1}$:$10^6$ to $10^3$:$10^6$. The data were expressed as mean ± SD, $n = 3$ independent replicates. ****$P < 0.0001$ $10^{-1}$:$10^6$ vs. 1:$10^6$, ****$P < 0.0001$ 1:$10^6$ vs. 10:$10^6$, ****$P < 0.0001$ 10:$10^6$ vs. $10^2$:$10^6$, ****$P < 0.0001$ $10^2$:$10^6$ vs. $10^3$:$10^6$, two-sided, multiple comparisons by One-way ANOVA test. $P < 0.05$ was considered statistically significant. Source data are provided as a Source Data file.

presence of both PGM and G6PDH, the concentration of NADPH increased within 0–10 min to reach a plateau of $129.21 \pm 3.68$ µM (Supplementary Fig. 19, purple curve). No NADPH was generated in the absence of either PGM (Supplementary Fig. 19, blue curve) or G6PDH (Supplementary Fig. 19, orange curve). These results confirmed the successful reconstitution of the glycogenolysis pathway in artificial cells.

### Artificial cells capable of signal transduction mediated by ADRB2 for the regulation of glycogenolytic metabolism

By encapsulating the glycogenolytic pathway into artificial cells capable of transducing extracellular primary messenger (ISO) into intracellular secondary messenger (cAMP), artificial cells capable of regulating glycogenolytic metabolism mediated by ADRB2 (Fig. 1a) were constructed. The artificial cells were reconstituted with ADRB2-Gsα complex and ADCY5 in the membrane and encapsulated with 1.0 U/mL PKA, 1.4 U/mL PhK, 1.0 U/mL PYGM, 1.0 U/mL PGM and 1.0 U/mL G6PDH in the lumen. With the addition of 1.0 µM ISO, the production of G-1-P inside artificial cells at a density of $10^7$ mL$^{-1}$ was confirmed by mass spectrometry (Supplementary Fig. 20), and was further quantified by HPLC (Fig. 6a, Supplementary Fig. 21). The concentration of produced G-1-P gradually increased from 0 to 25 min and then plateaued, with a maximum value of $48.88 \pm 2.01$ µM (Fig. 6a, purple curve). No G-1-P was observed without ISO stimulation (Fig. 6a, orange curve). NADPH production inside artificial cells at a density of $10^7$ mL$^{-1}$ was visualized by fluorescence microscopy (Fig. 6b). The blue fluorescence intensity inside the artificial cell gradually increased until 25 min, as demonstrated by the fluorescence spectroscopy data (Supplementary Fig. 22). The quantification of the produced NADPH concentration using UV–vis spectroscopy (Fig. 6c, blue curve, and Supplementary Fig. 23) revealed results similar to those for G-1-P. The NADPH concentration reached $122.24 \pm 10.39$ µM at 25 min. No NADPH was produced without the addition of ISO (Fig. 6c, red curve). All the abovementioned results confirmed the successful construction of artificial cells capable of regulating glycogenolytic metabolism through signal transduction from the primary messenger ISO to intracellular cAMP.

G-1-P production was regulated by the ISO concentration (Fig. 6d). At ISO concentrations of 0.01 and 0.1 µM, no G-1-P was produced because no cAMP was produced under these conditions (Fig. 4f). At ISO concentrations of 1.0, 10.0 and 100.0 µM, the concentration of G-1-P at 30 min inside artificial cells remained the same (Fig. 6d). NADPH production inside artificial cells at 30 min was visualized with fluorescence microscopy at various concentrations of ISO (Fig. 6e). No significant difference in blue intensity inside the artificial cells was observed at ISO concentrations of 1.0, 10.0 and 100.0 µM. A calibration curve was constructed using UV-Vis to quantify NADPH (Supplementary Fig. 23). The further quantitation of NADPH inside artificial cells after 30 min of stimulation revealed a similar pattern, with an NADPH

concentration of $121.63 \pm 10.91$ µM with 1.0 µM ISO (Fig. 6f). Quantitative analysis revealed progressive signal intensification, with $22.45 \pm 2.14$-fold amplification from ISO to cAMP, which increased to $48.88 \pm 2.01$-fold amplification at the G-1-P production stage and reached $121.63 \pm 10.91$-fold amplification at the 6-PGL generation phase (Fig. 6g). This hierarchical amplification is characteristic of GPCR-mediated signal transduction.

## Discussion

β2-Adrenergic receptor (ADRB2), Gs subunit α (Gsα) and activated adenylate cyclase (ADCY5) were purified and reconstituted into artificial cell membranes for signal transduction from the extracellular primary messenger ISO to the intracellular secondary messenger cAMP. cAMP production inside artificial cells was visualized via Epac1-cAMP FRET probes. The concentration of cAMP increased in an ISO dose-dependent manner, with a maximum value of $57.42 \pm 2.64$ µM at 6.0 µM. The amplification fold change was $22.45 \pm 2.14$ with stimulation by 1.0 µM ISO. An antagonist (alprenolol) and an inverse agonist (carazolol) that specifically bind to ADRB2 regulate the signal transduction process, which further proves the successful construction of artificial cells capable of signal transduction mediated by GPCRs. After further encapsulating the glycogenolytic pathway, artificial cells capable of signal transduction to regulate downstream metabolic pathways were successfully constructed. cAMP activated following the phosphorylation of PhK and PYGM to obtain p-PYGM, which catalyzed the decomposition of glycogen into G-1-P. G-1-P was converted into G-6-P catalyzed by PGM, which was subsequently converted into 6-PGL by G6PDH, accompanied by the production of NADPH. The production of NADPH inside the artificial cells was visualized by fluorescence microscopy and quantified using UV–vis spectrometer, reaching $121.63 \pm 10.91$ µM with 1.0 µM ISO simulation for 30 min. A progressive signal intensification phenomenon was observed, with $22.45 \pm 2.14$-fold amplification from the ISO to the cAMP, increasing to $48.88 \pm 2.01$-fold amplification for G-1-P production and reaching $121.63 \pm 10.91$-fold amplification for 6-PGL generation. These findings demonstrate the successful reconstitution of the GPCR-mediated signaling pathway for downstream metabolism regulation in artificial cells. To improve the autonomous properties of current artificial cells, an ATP regeneration module can be encapsulated. The proposed artificial cells lay the foundation for the development of autonomous artificial cells.

## Methods
### Materials
1, 2-dioleoyl-sn-glycero-3-phosphocholine (DOPC), 1, 2-dioleoyl-sn-glycero-3-phospho-(1′-rac-glycerol) (DOPG) were purchased from Avanti Polar Lipids (USA). Dioleoyl-sn-glycero-3-phosphoethanolamine-N-(7-nitro-2-1,3-benzoxadiazol-4-yl) (NBD-PE), Texas Red-DHPE (TR-DHPE), fluorescein-5-isothiocyanate (FITC), cyanine5 maleimide (Cy5 Mal), BODIPY guanosine 5′-O-(3-thiotriphosphate) sodium salt

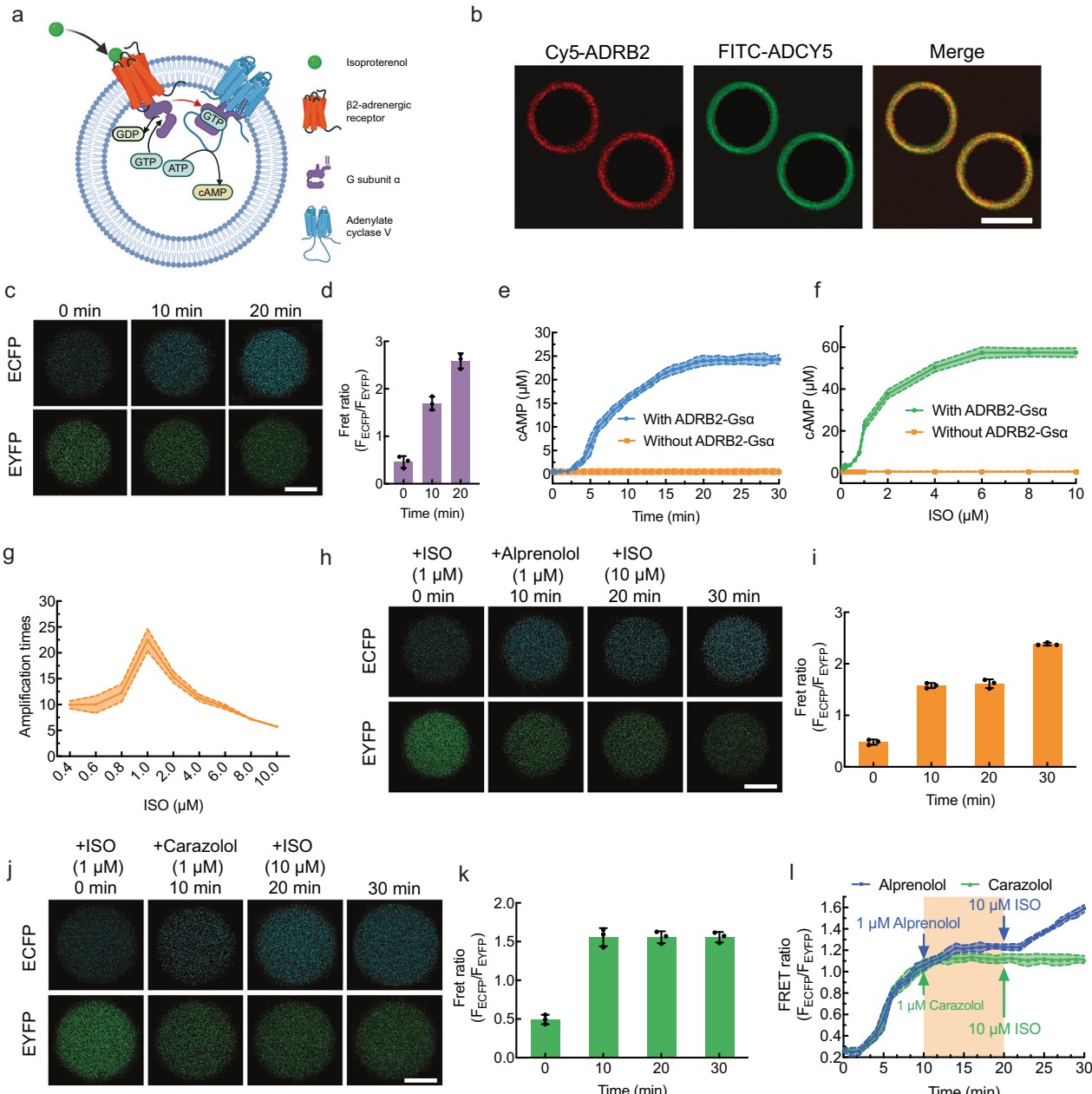

**Fig. 4 | Artificial cells capable of signal transduction from extracellular ISO to intracellular cAMP. a** Schematic illustration of the signal transduction of an artificial cell from extracellular ISO to intracellular cAMP. Created in BioRender. Liu, Y. (2025) https://BioRender.com/ik5i982. **b** Fluorescence images of GUVs containing Cy5-ADRB2-Gsα complexes and FITC-ADCY5 taken by red channel (left image), green channel (middle image), and their merged image (right image). The scale bar is 10.0 µm. *n* = 3 independent replicates. **c** The representative time-series images of artificial cells stimulated by 1.0 µM ISO for the intracellular cAMP production from 0 to 20 min, with top row images taken by the blue channel and bottom row images taken by the green channel. The scale bar is 10.0 µm. *n* = 3 independent replicates. **d** The corresponding FRET ratios of Epac1-cAMP in the artificial cell in (**c**). The data were expressed as mean ± SD, *n* = 3 independent replicates. **e** The cAMP production in artificial cells with the addition of 1.0 µM ISO as a function of time from 0 to 30 min using fluorescence spectroscopy. *n* = 3 independent replicates. **f** The cAMP production in artificial cells with the addition of varied ISO concentration (0.1, 0.2, 0.4, 0.6, 0.8, 1.0, 2.0, 4.0, 6.0, 8.0, 10.0 µM) at 30 min. *n* = 3 independent replicates. **g** The signal amplification folds of artificial cells from extracellular ISO to intracellular cAMP as a function of ISO concentration (0.4 to 10.0 µM) at 30 min. *n* = 3 independent replicates. **h** The representative time-series images of Epac1-cAMP in

artificial cells with sequential antagonist alprenolol treatment for 10 min from 10 to 20 min, and competitive 10.0 µM ISO activation for 10 min from 20 to 30 min, with top row images taken by blue channel and bottom row images taken by green channel. The scale bar is 10.0 µm. *n* = 3 independent replicates. **i** The corresponding Fret ratio of Epac1-cAMP in the artificial cell in (**h**). The data were expressed as mean ± SD, *n* = 3 independent replicates. **j** The representative time-series images of Epac1-cAMP in artificial cells with sequential inverse agonist carazolol treatment for 10 min from 10 to 20 min, and 10.0 µM ISO activation for 10 min from 20 to 30 min, with top row images taken by blue channel and bottom row images taken by green channel. The scale bar is 10.0 µm. *n* = 3 independent replicates. **k** The corresponding FRET ratio Epac1-cAMP in the artificial cell in (**j**). The data were expressed as mean ± SD, *n* = 3 independent replicates. **l** FRET ratio of Epac1-cAMP in the artificial cell measured by fluorescence spectroscopy as a function of time with the addition of 1.0 µM alprenolol antagonism at 10 min and the addition of 10.0 µM ISO at 20 min (blue curve), with the addition of 1.0 µM carazolol inverse agonism at 10 min and the addition of 10.0 µM ISO at 20 min (green curve). The data were expressed as mean ± SD, *n* = 3 independent replicates. Source data are provided as a Source Data file.

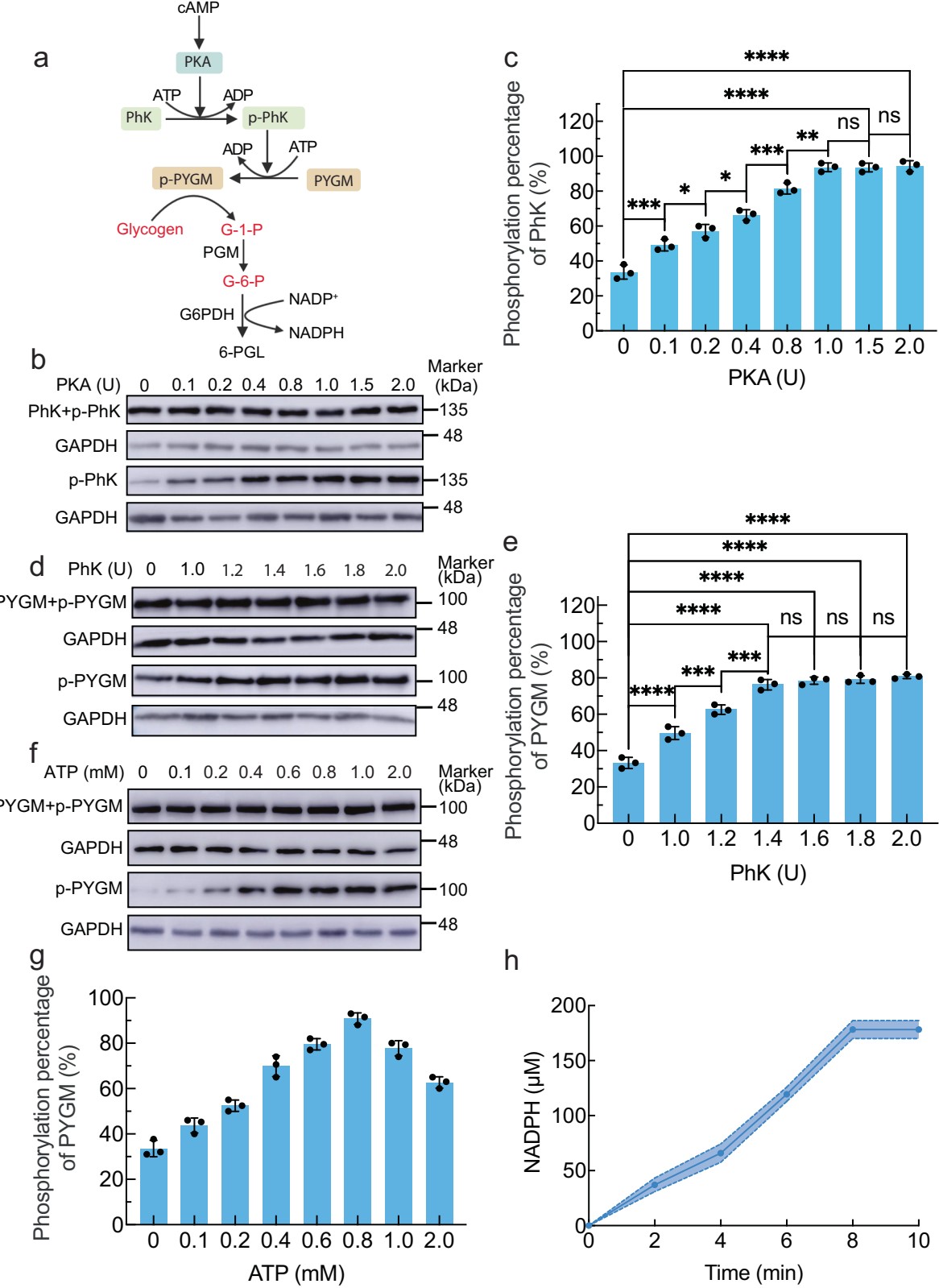

(BODIPY TR GTPγS), lauryl maltose neopentyl glycol (LMNG) and cellfectin reagent were purchased from Invitrogen (USA). Octyl β-D-glucopyranoside (OG), n-dodecyl-β-D-maltoside (DDM), guanosine 5′-triphosphate (GTP) sodium salt, guanosine 5′-diphosphate (GDP) sodium salt, adenosine 5′-triphosphate (ATP) disodium salt, and 3β-Hydroxy-5-cholestene 3-hemisuccinate (CHS) were purchased from Sigma-Aldrich (USA). Isoprenaline hydrochloride (ISO·HCl), carazolol

and alprenolol were purchased from MedChemExpress (China). Gentamicin, kanamycin, tetracycline, 2-[4-(2-hydroxyethyl)piperazin-1-yl] ethanesulfonic acid (HEPES), and 5-bromo-4-chloro-3-indolyl β-D-galactoside (X-gal) were purchased from Beyotime (China). Easy PAGE gel fast preparation kit (12.5%) was purchased from Seven (China). Bio-beads SM-2 were purchased from Bio-Rad (USA). Glycogen, cyclic adenosine monophosphate (cAMP, HPLC ≥ 98%) and α-D-

**Fig. 5 | Construction of glycogenolytic pathway. a** The scheme of the cAMP-regulated glycogenolytic pathway. **b** Western blot images of PhK+p-PhK (first row images) and phosphorylated PhK (third row images) with a variation of PKA from 0 to 2.0 U at fixed initial PhK of 1.0 U and 0.5 mM ATP. $n = 3$ independent replicates. The samples derive from the same experiment, and the blots were processed in parallel. **c** Relative phosphorylation percentage of PhK obtained from the corresponding intensity of WB bands of p-PhK over (PhK + p-PhK) in (**b**). The data were expressed as mean ± SD, $n = 3$ independent replicates. ***$P = 0.0005$ 0 vs. 0.1, *$P = 0.0462$ 0.1 vs. 0.2, *$P = 0.0496$ 0.2 vs. 0.4, ***$P = 0.0006$ 0.4 vs. 0.8, **$P = 0.0050$ 0.8 vs. 1.0, ns$P > 0.05$ 1.0 vs. 1.5, ns$P > 0.05$ 1.5 vs. 2.0, ****$P < 0.0001$ 0 vs. 1.5, ****$P < 0.0001$ 0 vs. 2.0, two-sided, multiple comparisons by One-way ANOVA test. **d** Western blot images of PYGM+p-PYGM (first row images) and phosphorylated PYGM (third row images) with a variation of PhK from 0 to 2.0 U at fixed initial PYGM of 1.0 U, PKA of 1.0 U, and ATP of 0.5 mM. $n = 3$ independent replicates. The samples derive from the same experiment, and the blots were processed in parallel. **e** Relative phosphorylation percentage of PYGM obtained from the corresponding intensity of WB bands of p-PYGM over (PYGM+p-PYGM) in (**d**). The data were expressed as mean ± SD, $n = 3$ independent replicates. ****$P < 0.0001$ 0 vs. 1.0, ***$P = 0.0004$ 1.0 vs. 1.2, ***$P = 0.0002$ 1.2 vs. 1.4, ns$P > 0.05$ 1.4 vs. 1.6, ns$P > 0.05$ 1.6 vs. 1.8, ns$P > 0.05$ 1.8 vs. 2.0, ****$P < 0.0001$ 0 vs. 1.4, ****$P < 0.0001$ 0 vs. 1.6, ****$P < 0.0001$ 0 vs. 1.8, ****$P < 0.0001$ 1.8 vs. 2.0, two-sided, multiple comparisons by One-way ANOVA test. **f** West blot images of PYGM+p-PYGM (first row images) and phosphorylated PYGM (third row images) with a variation of ATP concentration from 0 to 2.0 mM at fixed PKA of 1.0 U, PhK of 1.4 U and PYGM of 1.0 U. $n = 3$ independent replicates. The samples derive from the same experiment, and the blots were processed in parallel. **g** Relative phosphorylation percentage of PYGM obtained from the corresponding intensity of WB bands of p-PYGM over (PYGM+p-PYGM) in (**f**). The data were expressed as mean ± SD, $n = 3$ independent replicates. **h** The production of NADPH as a function of time in a solution containing 1.0 μM cAMP, 1.0 U PKA, 1.4 U PhK, 1.0 U PYGM, 0.8 mM ATP, 0.5 mM glycogen, 0.2 mM NADP+. The data were expressed as mean ± SD, $n = 3$ independent replicates. $P < 0.05$ was considered statistically significant. Source data are provided as a Source Data file.

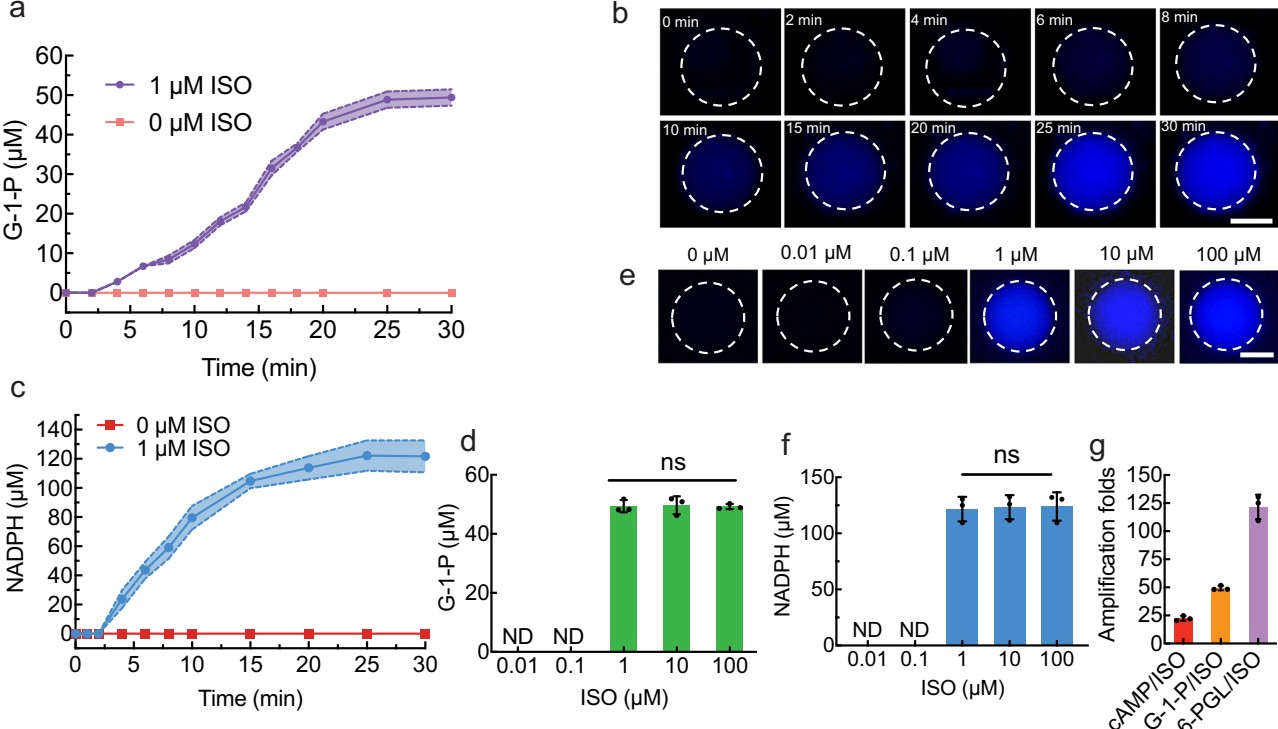

**Fig. 6 | The artificial cell capable of signal transduction mediated by ADRB2 for the regulation of glycogenolytic metabolism. a** The concentration of produced G-1-P as a function of time inside artificial cells containing (0.8 mM ATP, 0.5 mM glycogen, 0.2 mM NADP+, 1.0 U/mL PKA, 1.4 U/mL PhK, 1.0 U/mL PYGM, 1.0 U/mL PGM, 1.0 U/mL G6PDH) stimulated by 1 μM ISO. The data were expressed as mean ± SD, $n = 3$ independent replicates. **b** Representative fluorescence images of produced NADPH as a function of time with the addition of 1.0 μM ISO. The dashed white circles represented artificial cell membranes. The scale bar is 10.0 μm. $n = 3$ independent replicates. **c** The concentration of produced NADPH obtained from UV−Vis spectroscopy inside artificial cells stimulated by 1 μM ISO. The data were expressed as mean ± SD, $n = 3$ independent replicates. **d** The concentration of produced G-1-P inside artificial cells at 30 min with various ISO concentrations of 0.01, 0.1, 1.0, 10.0 and 100.0 μM. The data were expressed as mean ± SD, $n = 3$ independent replicates. ns$P > 0.05$, two-sided, multiple comparisons by One-way ANOVA test. **e** Representative fluorescence images of produced NADPH inside artificial cells at 30 min with various ISO concentrations of 0, 0.01, 0.1, 1.0, 10.0 and 100.0 μM. The dashed white circles represented artificial cell membranes. The scale bar is 10.0 μm. $n = 3$ independent replicates. **f** The concentration of produced NADPH inside artificial cells obtained from UV−Vis spectroscopy at 30 min with various ISO concentrations of 0.01, 0.1, 1.0, 10.0 and 100.0 μM. The data were expressed as mean ± SD, $n = 3$ independent replicates. ns$P > 0.05$, two-sided, multiple comparisons by One-way ANOVA test. **g** Signal amplification folds of cAMP, G-1-P, and 6-PGL to ISO of the whole signal transduction pathway from extracellular ISO to 6-PGL in the artificial cells. The data were expressed as mean ± SD, $n = 3$ independent replicates. Source data are provided as a Source Data file.

Glucose-1-phosphate (G-1-P, Biotech, 98%) were purchased from Yuanye (China). Sodium chloride (NaCl), hydrochloric acid (HCl), calcium chloride ($CaCl_2$), magnesium chloride ($MgCl_2$), sodium dihydrogen phosphate ($NaH_2PO_4$), potassium phosphate monobasic ($KH_2PO_4$) and gluconate potassium (K-gluconate) were purchased from Aladdin (China). Rabbit anti-PhKA2 polyclonal antibody (No. 24658-1-AP), rabbit anti-PYGM-Specific polyclonal antibody (No. 19716-1-AP) and ADRB2 polyclonal antibody (No. 29864-1-AP) were purchased from Proteintech (China). Anti-PYGL (phospho S430) + PYGM (phospho S430) antibody (No. EPR20852-26) was purchased from Abcam (China). Rabbit anti-phospho-Serine polyclonal antibody (No. bs-11993R) and Rabbit GAPDH polyclonal antibody (No. bs-10900R) were

purchased from Bioss (China). HRP-labeled goat anti-rabbit IgG (H + L) (No. A0208) was purchased from Beyotime (China). All primary antibodies were diluted at 1:1500 (v/v), and the secondary antibody was diluted at 1:3000 (v/v). DH10 Bac and BL21(DE3) CodonPlus-RIL competent cells were purchased from WEIDI (China). Sf9 cell line (No. CL-0205) was purchased from Procell Life Science & Technology (China) and was authenticated by multiplex amplification for species identification. ESF 921 insect cell culture medium was purchased from Expression Systems (USA). Fetal bovine serum was purchased from Gibco (USA). Millipore Milli-Q water with a resistivity of 18.2 MΩ•cm was used in the experiments.

## ADRB2, Gsα, ADCY5 purification

An N-terminally fused Flag and C-terminally fused 10His β2-adrenergic receptor (ADRB2) construct was cloned into the pFastBac1 vector (Supplementary Fig. 1, Supplementary Data 1). A C-terminally fused 6His Gs subunit α (Gsα) construct was cloned into the pFastBac1 vector (Supplementary Fig. 2, Supplementary Data 1). An N-terminally fused Flag adenylate cyclase V (ADCY5) construct was cloned into the pFastBac1 vector (Supplementary Fig. 3, Supplementary Data 1). The constructs were expressed in Sf9 insect cell cultures infected with recombinant baculovirus (Bac-to-Bac Baculovirus Expression System), which was solubilized using n-dodecyl-β-d-maltoside (DDM). The first purification of the ADRB2 receptor was performed using Anti-FLAG affinity resin, followed by the second purification using Ni-NTA agarose resin according to the recommended protocol. Gsα was purified by Ni-NTA agarose resin. ADCY5 was purified by Anti-FLAG affinity resin. Protein purity was assessed through SDS-PAGE analysis employing a standardized 12% (w/v) resolving gel under established electrophoretic conditions, while quantitative determination of protein concentrations was conducted using a colorimetric Bradford assay with BSA as the calibration standard.

## ADRB2-Gsα complex formation and purification

The stable ADRB2-Gsα complex was formed by mixing ISO-bound ADRB2 (130 μM) with Gsα (100 μM) in 2.0 ml of buffer (10.0 mM HEPES, pH 7.5, 100. mM NaCl, 0.1% DDM, 1.0 mM EDTA, 3.0 mM MgCl$_2$, 10.0 μM ISO) at room temperature for 3 h. To replace DDM combined with ADRB2-Gsα complex with LMNG, the ADRB2-Gsα mixture (2.0 ml) was mixed with 8.0 ml of exchange buffer (20.0 mM HEPES, pH 7.5, 100.0 mM NaCl, 10.0 μM ISO, 1.25% LMNG) at room temperature for 1 h. The intact ADRB2-Gsα complexes were obtained using a Superdex 200 10/300 GL column (GE Healthcare) by removing free ADRB2 and free Gsα.

## Assay of ADCY5 activity

The activity of ADCY5 was measured at 37 °C by monitoring the increase of cAMP using high-performance liquid chromatography (HPLC). HPLC with a C18 column (4.0 × 250 mm, 30 °C) linked to a refractive-index detector (UV detector at 254 nm). For the determination of cAMP, reaction samples (5.0 μL) were injected and eluted with 0.05 M KH$_2$PO$_4$ (pH 4) – acetonitrile (90:10) at a flow rate of 1.0 mL/min. The concentration of cAMP was determined using the calibration curve (Supplementary Fig. 4h). The assay mixture contained ADCY5 (6.0 μg/mL), ADRB2-Gsα (6.0 μg/mL) and 10.0 μM ISO in 50.0 mM HEPES buffer (pH 7.4). Enzyme kinetic analysis was performed with various ATP concentrations (from 0 to 14.0 mM). The $V_{max}$ and $K_m$ values were calculated by fitting the initial-rate data to the Michaelis–Menten equation with a nonlinear regression analysis program.

The optimal pH of ADCY5 was measured in reaction buffer (50.0 mM HEPES pH 7.4, 6.0 μg/mL ADCY5, 6.0 μg/mL ADRB2-Gsα, 10.0 μM ISO and 0.8 mM ATP) under the assay conditions mentioned above. The optimal temperature of ADCY5 was measured at its optimal pH. The optimal Mg$^{2+}$ concentration was measured at its optimal pH and temperature.

## Construction of artificial cells capable of signal transduction mediated by ADRB2

The artificial cells were prepared using a gel-assisted hydration method[34]. PVA (0.2 g) was dissolved in pure water (10 mL) by stirring at 90 °C for 2 h. The PVA solution was filtered through a 0.22 μm hydrophilic polyethersulfone (PES) membrane to remove insoluble aggregates. PVA film was formed on the bottom of a glass vial by evaporating 100.0 μL PVA solution (0.02 g/mL) at 50 °C for 4 h. Subsequently, 10.0 μL of phospholipid chloroform solution (0.9 mg/mL DOPC, 0.1 mg/mL DOPG, and 0.05 mg/mL CHS) was evenly spread onto the dried PVA film. The vial was then subjected to vacuum desiccation at 25 °C for 2 h to remove residual organic solvent to form a dry lipid film. Reconstitution buffer containing ADRB2 (20.0 mM HEPES pH 7.4, 100.0 mM NaCl, 0.8 mM ATP, 2.0 mM MgCl$_2$, 300.0 mM sucrose) was added into the abovementioned vial with a lipid-to-ADRB2 molar ratio ranging from 10$^2$:1 to 10$^6$:1. To remove LMNG bound to ADRB2, pretreated Bio-Beads SM-2 were added to the solution at a mass ratio of 1:5 (beads: buffer). The mixture was gently shaken at 30 rpm on a shaker at 4 °C for 2 h. The beads were then removed by centrifugation at 200 × $g$ for 5 min. The resulting artificial cells were collected and washed three times with the buffer (20.0 mM HEPES, pH 7.4, 100.0 mM NaCl, 0.8 mM ATP, 2.0 mM MgCl$_2$, 300.0 mM glucose) via centrifugation at 1000 × $g$ for 5 min at 4 °C to remove non-reconstituted proteins. A similar protocol was applied to fabricate artificial cells reconstituted with ADRB2-Gsα, ADCY5, and ADRB2-Gsα + ADCY5, respectively.

To control the size of artificial cells, the artificial cells were further extruded through polycarbonate filter membranes with pore sizes of 3 μm or 10 μm three times, respectively. The size distribution of the artificial cells was determined by analyzing microscopy images of at least 300 individual GUVs using ImageJ software. The density of artificial cells was directly quantified using an automated cell counter, by adding artificial cell solution (10 μL) to the cell counting chamber.

## Flow cytometry analysis

GUVs without membrane proteins served as blank controls. The 660/20 nm bandpass filter and 525/40 nm bandpass filter were used for Cy5 labeled ADRB2-Gsα detection and FITC labeled ADCY5 detection, respectively. The flow rate was set to 10 μL/min to ensure single GUV detection. The threshold was adjusted based on forward scatter (FSC) and side scatter (SSC) to exclude small lipid aggregates, with only events with FSC-A > 10$^4$ and SSC-A > 10$^3$ to be GUVs. The flow cytometry data acquisition for each sample included 10$^4$ GUVs. GUVs without membrane proteins were analyzed to establish the autofluorescence background, with a threshold to be 99% of this population. GUVs with membrane proteins exhibiting fluorescence intensity above this threshold were identified as positive samples.

## Assay of the percentage of GUVs containing ADRB2-Gsα complex and ADCY5

Flow cytometry was used for determining the percentage of GUVs containing ADRB2-Gsα complex and ADCY5. The percentage of GUVs containing membrane proteins was calculated using the following Eq. (1):

$$P = \frac{N_{positive}}{N_{total}} \times 100\% \tag{1}$$

where $N_{positive}$ is the number of GUVs containing membrane proteins, $N_{total}$ is the number of total GUVs.

## Calculation of the number of ADRB2 per artificial cell

The number of ADRB2 in each artificial cell was calculated by following Eq. (2)[22], assuming no loss of ADRB2:

$$N_{ADRB2} = N_{lipids} \times \frac{receptor}{lipid} ratio \qquad (2)$$

where $N_{ADRB2}$ is the number of ADRB2 per artificial cell, $N_{lipids}$ is the number of lipids per artificial cell. The number of lipids per artificial cell was calculated by its surface area over the cross-area of the DOPC lipid molecule (70 Å$^2$), multiplying by 2 due to the bilayer structure of the cell membrane. The artificial cell surface area was $6.16 \times 10^{10}$ Å$^2$ due to its diameter of $14.0 \pm 3.0$ μm. The number of lipids per cell was estimated to be $1.76 \times 10^9$ lipids; thus, the amount of ADRB2 per artificial cell was $1.8 \times 10^6$ ADRB2 per cell, with the lipid-to-ADRB2 molar ratio of 1000:1.

## Assay of the ADRB2-Gsα complexes reconstitution

The artificial cells reconstituted with ADRB2-Gsα complexes were obtained using a similar method to ADRB2 reconstituted artificial cells, by replacing ADRB2 with ADRB2-Gsα. The reconstitution of ADRB2-Gsα complexes was visualized by BODIPY TR GTPγS, which bound to Gsα to produce red fluorescence. Following the addition of 1 μM ISO outside artificial cells, their fluorescence images were taken using a laser scanning confocal microscopy with red fluorescence at 610 nm. The fluorescence intensity of BODIPY TR GTPγS was recorded by a fluorescence spectrometer ($\lambda_{ex} = 590$ nm, $\lambda_{em} = 610$ nm).

## Expression and purification of Epac1-cAMP probe

The Epac1-cAMP cDNA containing a C-terminal 6His tag was inserted into the pET11a expression vector (Supplementary Data 1). Protein production occurred in BL21(DE3) CodonPlus-RIL *E. coli* cells induced with 1 mM IPTG, followed by cultivation at 37 °C for 4 h and subsequent overnight incubation at 18 °C. Bacterial pellets were suspended in NPI-20 buffer (50.0 mM NaH$_2$PO$_4$, 300.0 mM NaCl, 20.0 mM imidazole, pH 7.4) containing 1 mg/ml lysozyme and DNase I, then incubated on ice for 30 min. Following cell lysis with 10 cycles of 15-second sonication at 40% amplitude, the Epac1-cAMPs protein was purified using Ni-NTA affinity chromatography with 5 column volumes of elution buffer (50.0 mM NaH$_2$PO$_4$, 300.0 mM NaCl, 150.0 mM imidazole, pH 7.4).

## Assay of cAMP in artificial cells

The cAMP produced inside the artificial cells was measured with the Epac1-cAMP probe. The artificial cells were reconstituted with ADRB2-Gsα and ADCY5 and with the lumen solution of 20.0 mM HEPES pH 8, 300.0 mM sucrose, 100.0 mM NaCl, 0.5 mM ATP, 2.0 mM MgCl$_2$. To visualize the generation of cAMP in individual artificial cells in real-time, the Epac1-cAMP probe (0.7 μM) was encapsulated within the lumen of GUVs. Time-lapse images were acquired. The GUVs were observed using a laser scanning confocal microscope at 37 °C with two channels of ECFP and EYFP excited by a 405 nm laser. The quantitative measurement of cAMP concentration was performed using fluorescence spectroscopy. Following the addition of various concentrations of ISO outside of the artificial cells for 30 min, 10% Triton X-100 was introduced to disrupt the artificial membranes, with subsequent addition of 5% trichloroacetic acid (12% w/v) to halt reactions. 10 μL Epac1-cAMP (0.7 μM, 135.0 mM K-gluconate, 12.0 mM NaHCO$_3$, 4.0 mM KCl, 0.8 mM MgCl$_2$, 10.0 mM HEPES, pH 7.4) was mixed with 90.0 μL of lysed artificial cell solution to measure the fluorescence intensity of Epac1-cAMP probes with excitation at 430 nm and emission detection at 475 nm (ECFP) and 530 nm (EYFP). The cAMP concentration was determined using the calibration curve (Supplementary Fig. 12).

## Assay of phosphorylation of the enzymes involved in glycogenolytic pathway

The glycogenolytic pathway involved PKA, PhK and PYGM. Their phosphorylation was analyzed via Western blot with their corresponding antibodies. Enzymes were incubated in the reaction buffer (20.0 mM HEPES, pH 8, 100.0 mM NaCl, 0.5 mM ATP, 2.0 mM MgCl$_2$, 1.0 mM CaCl$_2$) for 15 min, followed by heating at 80 °C for 10 min to terminate the reaction. Anti-PhKA2 and anti-PYGM specific antibodies were used to recognize PhK and PYGM, respectively. Anti-phospho-serine and anti-PYGL (phospho S430) + PYGM (phospho S430) were used to recognize phosphorylated PhK and PYGM, respectively. All primary antibodies were diluted at 1:1500 (v/v) in blocking buffer. The images of those WB bands were taken using the chemiluminescence mode of a gel imaging system.

## HPLC assay of G-1-P

G-1-P lacks a strong chromophore for sensitive UV detection. Therefore, prior to HPLC analysis, G-1-P underwent aniline derivatization. The lysed artificial cell solution was filtrated through a 0.22 μm Nylon 66 membrane to remove proteins. Afterwards, a 200 μL aliquot was mixed with 0.2 M aniline hydrochloride solution (100 μL, pH 2.0). The mixture was incubated at 60 °C for 30 min to allow the Schiff base formation between the aldehyde group of ring-opened G-1-P and the amine group of aniline. The reaction was then quenched by adding 1.0 M NaOH (100 μL). To stabilize the derivative, 0.2 M sodium cyanoborohydride (NaBH$_3$CN) (200 μL) was added at room temperature for 1 h to reduce the Schiff base to a stable secondary amine. The excess aniline was removed by extracting the mixture with 500 μL of dichloromethane (DCM). The upper aqueous layer containing the derivatized G-1-P was collected, followed by filtering through a 0.22 μm Nylon 66 membrane. Subsequently, 5.0 μL aliquots were injected into the HPLC instrument equipped with a reverse-phase C18 column (4.6 × 250 mm, 5 μm particle size) at 30 °C. H$_2$O−acetonitrile (90:10, v/v) was used as elution with a flow rate of 1 mL/min. UV absorbance was monitored at 280 nm. The G-1-P concentration was obtained using the calibration curve (Supplementary Fig. 21h).

## LC-MS assay of G-1-P

With the addition of ISO (1 μM) outside of artificial cells for 30 min, 10% Triton X-100 was added to lyse artificial cells, followed by adding 5% trichloroacetic acid (12% w/v) to terminate reactions. The sample ($n = 3$ independent replicates) was centrifuged at 4 °C and 12,000 × $g$ for 15 min to collect the supernatant, which was filtered through a 0.22 μm Nylon 66 membrane before injection. Chromatographic separation was performed on a BEH Amide column (2.1 mm × 100 mm, 1.7 μm) at 35 °C. Mobile phase A consisted of a 5 mM ammonium acetate aqueous solution (pH 4.5), and mobile phase B was acetonitrile. The gradient elution program was set as follows: 0−4 min, linear decrease of phase B from 90% to 70%; 4−4.1 min, increase of phase B to 90%; and 4.1−6 min, re-equilibration at 90% of phase B, with a flow rate of 0.3 mL·min$^{-1}$. Mass spectrometric detection was carried out using an electrospray ionization source in negative ion mode, with the spray voltage of −3.2 kV, the ion source temperature of 330 °C, the sheath gas pressure of 35 psi, the auxiliary gas pressure of 8 psi, and the collision gas (argon) pressure of 1.2 mTorr. The precursor ion (theoretical m/z = 259.0220, [M−H]$^-$) was acquired in full-scan mode, and characteristic fragment ions were obtained via MS/MS.

## Assay of NADPH in the artificial cells reconstituted with ADRB2-Gsα and ADCY5 and encapsulated with glycogenolytic pathway

The artificial cells reconstituted with ADRB2-Gsα and ADCY5 and encapsulated with glycogenolysis pathway were constructed with the lumen solution containing 20 mM HEPES pH 8, 300 mM sucrose, 100.0 mM NaCl, 0.8 mM ATP, 2.0 mM MgCl$_2$, 0.5 mM glycogen,

0.2 mM NADP⁺, 1.0 U/mL PKA, 1.4 U/mL PhK, 1.0 U/mL PYGM, 1.0 U/mL PGM, 1.0 U/mL G6PDH. The fluorescence microscopy was used for qualitative visualization of NADPH production within artificial cells. At designated time points after the addition of ISO, 10 μL of the artificial cell solution was placed on a glass slide and immediately observed using an inverted fluorescence microscope with a blue channel. The quantitative determination of NADPH concentration was performed using UV–Vis spectroscopy. To terminate the signaling and metabolic reactions, the artificial cell solution was treated with 10% (v/v) Triton X-100 to release intracellular contents, followed immediately by the addition of trichloroacetic acid (TCA) to a final concentration of 5% (w/v) to denature and precipitate proteins. The resulting lysate was centrifuged at $10,000 \times g$ for 5 min at 4 °C, followed by carefully collecting the supernatant and filtering it through a 0.22 μm nylon 66 membrane. The concentration of NADPH in the supernatant was quantified by measuring its absorbance at 340 nm using a UV-vis spectrophotometer. The NADPH concentration was determined by the calibration curve (Supplementary Fig. 23).

## Characterizations

Laser scanning confocal microscope (Olympus FV 3000, Japan) and an inverted fluorescence microscope (Olympus IX73, Japan) were used to obtain all fluorescence images. A microplate reader (Molecular Devices, SpectraMax iD3, Germany) was used to measure the protein concentration. An ultrasonic homogenizer (Scientz, JY92-IIN, China) was used to lyse cells. A Cary 60 UV–vis spectrophotometer (Agilent, USA) was used for UV–vis spectra. A fluorescence spectrometry (PerkinElmer LS55, USA) was used for photoluminescence spectroscopy measurements. Amersham Imager 600 was used to capture the protein bands (GE HealthCare, USA). HPLC analysis was performed on an UltiMate 3000 HPLC system (Thermo Fisher Scientific, USA). Flow cytometry was performed on a flow cytometer (BD FACS Aria Fusion, USA). LC-MS was performed on a liquid chromatography-mass spectrometry (Agilent 1290II-6545, USA). An automated cell counter (Countess II, Thermo Fisher Scientific, USA) was used to characterize the density of artificial cells.

## Statistics & reproducibility

The quantitative values were presented as the mean ± standard deviation for the data. Sample size (n) for each statistical analysis was specified in the figure legends. The significance of differences was analyzed using a two-sided, multiple comparisons by one-way ANOVA. Statistical significance was set at $P < 0.05$. No statistical method was used to predetermine sample size. No data were excluded from the analyses. All statistical analyses were performed using GraphPad Prism 10.0. The intensities of fluorescence images and WB bands were analyzed using ImageJ 1.8.0.

## Reporting summary

Further information on research design is available in the Nature Portfolio Reporting Summary linked to this article.

## Data availability

All data generated in this study are provided in the Supplementary Information and Source Data files. The nucleotide sequence of ADRB2 is available at the NCBI accession NM_000024 [https://www.ncbi.nlm.nih.gov/nuccore/NM_000024.6]. The nucleotide sequence of Gsα is available at the NCBI accession NM_000516 [https://www.ncbi.nlm.nih.gov/nuccore/NM_000516.7]. The nucleotide sequence of ADCY5 is available at the NCBI accession NM_001199642 [https://www.ncbi.nlm.nih.gov/nuccore/NM_001199642.1]. The nucleotide sequence of Epac1-cAMP is available at the NCBI accession NM_165490 [https://www.ncbi.nlm.nih.gov/nuccore/NM_165490.2]. Source data are provided with this paper.

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

## Acknowledgments

This work was supported by the National Natural Science Foundation of China (Nos. 22374033 and 22174031 to X.J.H.), and the Natural Science Foundation of Heilongjiang Province (No. ZD2022B001 to X.J.H.).

## Author contributions

Conceptualization: Y.H.L and X.J.H. Investigation: Y.H.L, W.Z, Y.M.Z, S.B.L, and R.Y.S. Visualization: Y.H.L, W.Z, X.X.Z, and J.J.Z. Funding acquisition: X.J.H. Project administration: X.J.H and S.B.L. Supervision: X.J.H. Writing – original draft: Y.H.L., S.B.L., and X.J.H. Writing – review & editing: Y.H.L and X.J.H.

## Competing interests

The authors declare no competing interests.
