## [Peer Review File · Nature Communications]

An artificial cell capable of signal transduction mediated by ADRB2 for the regulation of glycogenolysis

Corresponding Author: Prof Xiaojun Han

Version 0:

Reviewer comments:

Reviewer #1

(Remarks to the Author)

The authors report on the bottom-up construction of artificial cells capable of transducing isoproterenol (ISO) signals via adrenergic receptors to achieve spatiotemporal regulation of glycogenolytic metabolism. By incorporating natural G protein-coupled receptors (GPCRs) into lipid-based artificial cells, they successfully reconstituted the native GPCR signaling pathway. This was confirmed by introducing extracellular messenger ISO which led to production of cAMP inside artificial cells, verified using Epac1-cAMP FRET probes. The glycogenolytic pathway was also additionally incorporated within the aqueous lumen of these artificial cells to validate the regulation of downstream metabolic pathways.

The overall concept seems novel as it further expands the implications of artificial cell platform in reconstituting the natural complex cellular signaling pathway beyond simply incorporating synthetic channels with limited physiological relevancy. Also, the manuscript is well-structured and easy to follow. However, there are several aspects regarding the fabrication method as well as the physicochemical property of the resulting artificial cells prepared that need to be clarified and discussed more in detail to fully appreciate the novelty and scientific rigor of the work. Nevertheless, I believe that this study represents a significant advance in the field of artificial cell research, and the work is timely and thus would be of general interest to the researchers working in the relevant scientific community. Therefore, I recommend publication of this work in Nature Communication and hope that the following suggestions and comments help improve the manuscript.

1. While the abstract emphasizes the "autonomous" nature of the platform, the discussion would benefit from a clearer explanation of how this autonomy is achieved in practice. For instance, in the case of ATP, which is commonly used both upstream and downstream, wouldn't its continuous consumption through sustained ISO influx potentially lead to depletion?
2. Figure 1 would benefit from overall improvement in quality. Several abbreviations used in the main text do not align with the presented figure, requiring back-and-forth referencing, thereby reducing the readability. In addition, the separation between cAMP production and the downstream glycogenolytic pathway is not clearly delineated, making it difficult to grasp the conceptual flow at a glance. It is recommended to consider moving the abbreviations to the figure caption and reorganizing the layout to enhance clarity and logical structure.
3. The presentation of the system's behavior in solution followed by its application in GUV membranes facilitates understanding. However, it also raises the question of why membrane incorporation is essential. Providing insights into the native permeability of the GUV membrane, or any related characterization data, could further support the rationale and strengthen the overall narrative.
4. Diffusion coefficient of ADRB2 in the GUV appears to be higher than that in the native cell membrane. It would be important to provide additional experiments or, at the very least, a discussion to address the underlying reasons for this difference and how it may influence signal transduction dynamics.
5. It is quite remarkable that both ADRB2-Gsal complexes and ADCY5 are incorporated into artificial cells with such high yield. Is this expected? If so, it might be better to provide some detail on how these are experimentally incorporated into the GUV.
6. In line 267, ...N-terminal "enhanced" yellow fluorescent protein? Is this correct?
7. Why does the phosphorylation percentage of PYGM decrease after 0.8 mM of ATP?
8. How does the size of artificial cells influence the experimental outcomes? Additionally, it would be helpful to clarify whether and how the size was controlled or standardized during the experiments.
9. Lastly, it would be helpful if the authors could further emphasize the significance of reconstructing the glycogenolytic pathway starting from cAMP. Clarifying both the rationale behind selecting this specific signaling cascade and the evidence

supporting the successful reconstitution would strengthen the overall impact and relevance of the study.

Reviewer #2

(Remarks to the Author)

The manuscript by Liu et al presents the development of artificial cells equipped with a GPCR for transmembrane signaling and with a 5-enzyme intracellular signaling cascade. This study is presented in the context of “bottom up engineering of artificial cells”. As the manuscript states in the opening, artificial cells are designed to understand the working mechanism of mammalian cells. Transmembrane signaling in synthetic cells has indeed been a long standing challenge and as such the subject of this manuscript is indeed very important.

I find the manuscript very interesting but the level of presentation is disappointing. I cannot recommend this manuscript for publication.

1. One critique I have is that the manuscript is written in a “broken” English and in far too many instances, the sentences are grammatically incorrect. One can guess the overall meaning, but an article should not leave room for guessing.

2. The second major point of criticism is that the manuscript lacks a clear mission statement, an explanation of exactly what are the challenges that the manuscript addresses?

What has been done in the field, what were the failures that this manuscript overcomes ?

I find that the introduction does not present the state of art, does not explain the challenges, and does not present the prior steps taken to address these challenges. As such, the experimental work is presented as if in a vacuum of knowledge on the subject.

In my experience, the transfer of GPCR from mammalian to artificial cells has been a major challenge. But as presented, the manuscript does not explain, what has been the challenge ? what proteins have been reconstituted within artificial cells ? what methods are there to do this ?

3. There is a significant mis-match between the ambition of the manuscript to be published in Nature Communications and the data presentation: It is presented as if the GPCR signaling pathway is transplanted into an artificial cell and it works via routine protocols, without much optimization, it just does. So why is this worth Nature Communications if it does not address a challenge and is so easy ?

4. Introduction is not matched to the manuscript. The whole work is about reconstituting proteins. For this reason, I find that the introduction about artificial ion channels is redundant, because (to my knowledge) none of these examples of transmembrane signaling resulted in a biologically – relevant response within the artificial cell.

Subsequent paragraph on RTK and GPCR mimics is not incorrect but misses some critical references in which biologically relevant response has been achieved.

Results and discussion :

5. The manuscript as presented does not present sufficient level of experimental details to reproduce this work. As such, the process of creating GUVs, the incorporation of transmembrane proteins and that for the intracellular proteins are discussed hardly at all. With all the experience of my lab in this field, we would not be able to walk to the lab and reproduce this work. starting with the formation of GUV.

6. GPCR is a transmembrane protein, with seven transmembrane domains; Gs-a is a bilayer bound protein. How is it possible that the ADBR2-Gsa-ADCY5 connection is realized in solution ?

7. The correct GPCR orientation (over 94% yield !!!) is also presented as if it happens without optimization. This paragraph (lines 187-199) is really sketchy and hard if possible at all to follow.

8. The assay in Figure 3D is rather critical but is presented in details that leave too much out and is impossible to understand.

9. p.15, production of cAMP is measured as density per mL. What sort of system of measurements is this ?

10. if the connection between the agonist and cAMP is possible in solution, what benefit does an artificial cell have ? why bother making an artificial cell ?

11. the “final” composition of the GUV includes 3 proteins for transmembrane signaling and 5 for downstream signaling, each possibly with its own cofactors, necessary ions, starting materials, and then products formed. Some of these

components are membrane permeable and some are not – has this been controlled, quantified, analyzed? The experience of my lab is that these multicomponent mixtures are incredibly difficult to control and handle. Others report manuscript-long optimization of these protocols (see Danelon et al, Nature Comm) This manuscript presents it as a routine protocol.

12. Figure 6 : how was G6P quantified vis MS and HPLC ? there is no details on these experiments.

13. How were the artificial cells characterized for size and for concentration ?

14. Production of NADPH was monitored: as artificial cells, or with lysis ? again, little if any experimental details are provided.

15. How did you go from fluorescence to concentration within the artificial cell ? this is far from trivial.

16. The produced NADPH is reported in hundred micromolar concentration. This is an incredibly high concentration. I would like to see explicitly how was this number (and all the reported numbers) arrived at, because this concentration is remarkably high.

17. It is admirable that the GUV were analyzed via flow cytometry (fig 3h); these data illustrate that fluorescence can be different by as much as 100 fold. I do not understand how was the data in Figure 3i derived, and how are the error bars so small ?

18. It would be most appropriate to show microscopy images that with more than one GUV in the field of view;

19. The natural potency of agonists is typically nanomolar. The fact that all the GPCR activation in this work is conducted using micromolar ISO indicates that the GPCR is not in its native conformation, or is otherwise non-functional.

20. The manuscript critically misses on the explanation of the motivation of this work. Reconstituting the natural components does not reveal any new information about this reaction network. The proposed synthetic cell is simpler but what information does it reveal or make possible to arrive at ?

21. Finally, last but not least :

This journal pays close attention to statistical evaluation of data, the reproducibility of work. The data presented throughout the manuscript have remarkably small error bars. Remarkably small.

I would like the authors to clearly state if experiments presented in this work have been reproduced in INDEPENDENT experiments ? as disclosed, does n=3 refer to technical replicates or independent experiments ? if this has not been done in truly independent experiments, it has to be done so.

On the same subject : to my knowledge, t-test is not an appropriate statistical method here and all the data have to be analyzed via ANOVA.

Reviewer #3

(Remarks to the Author)

Version 1:

Reviewer comments:

Reviewer #1

(Remarks to the Author)

The authors have addressed all the comments and issues that I have raised and I am satisfied with these revisions. Therefore, I recommend publication without further revisions.

Reviewer #4

(Remarks to the Author)

I have carefully examined the authors' point-by-point responses to Reviewer #2 as well as the corresponding revisions in the manuscript. In my evaluation, the authors have made substantial and appropriate efforts to address the previously raised concerns. They have provided detailed experimental procedures for GUV construction, protein incorporation, and downstream metabolic assays; clarified methodological ambiguities; and added missing analytical details. The quality of the figures has also been improved, and additional experiments have been performed to support key conclusions, particularly regarding signal transduction efficiency and downstream metabolic outputs.

Overall, the technical issues highlighted by Reviewer #2 have been satisfactorily resolved. I believe that the revised manuscript presents a coherent and technically sound study, and in my opinion, it is suitable for publication in Nature Communications in its current form.

For the sake of clarity, the comments of the reviewer have been collated in black, and our response to each comment appears in blue. All the changes to the manuscript are highlighted in red.

Reviewer #1 (Remarks to the Author):

The authors report on the bottom-up construction of artificial cells capable of transducing isoproterenol (ISO) signals via adrenergic receptors to achieve spatiotemporal regulation of glycogenolytic metabolism. By incorporating natural G protein-coupled receptors (GPCRs) into lipid-based artificial cells, they successfully reconstituted the native GPCR signaling pathway. This was confirmed by introducing extracellular messenger ISO which led to production of cAMP inside artificial cells, verified using Epac1-cAMP FRET probes. The glycogenolytic pathway was also additionally incorporated within the aqueous lumen of these artificial cells to validate the regulation of downstream metabolic pathways.

The overall concept seems novel as it further expands the implications of artificial cell platform in reconstituting the natural complex cellular signaling pathway beyond simply incorporating synthetic channels with limited physiological relevancy. Also, the manuscript is well-structured and easy to follow. However, there are several aspects regarding the fabrication method as well as the physicochemical property of the resulting artificial cells prepared that need to be clarified and discussed more in detail to fully appreciate the novelty and scientific rigor of the work. Nevertheless, I believe that this study represents a significant advance in the field of artificial cell research, and the work is timely and thus would be of general interest to the researchers working in the relevant scientific community. Therefore, I recommend publication of this work in Nature Communication and hope that the following suggestions and comments help improve the manuscript.

Comments:

1. While the abstract emphasizes the "autonomous" nature of the platform, the discussion would benefit from a clearer explanation of how this autonomy is achieved in practice. For instance, in the case of ATP, which is commonly used both upstream and downstream, wouldn't its continuous consumption through sustained ISO influx potentially lead to depletion?

Thank the reviewer for the comments. The ultimate goal of artificial cells is to build an autonomous artificial cell, which can self-reproduce via the communication with its environments in terms of materials, energy and signals. Our work contributes to this field by enabling the artificial cell to sense the signal from environment, which is the emphasis of this work. We fully agree that ATP was continuously consumed and not continuously self-supplied in current work. This issue can be solved by involving photosynthetic organelles for ATP sustained supply. We added below sentence in page 25.

[...] These findings demonstrate the successful reconstitution of the GPCR-mediated signalling pathway for downstream metabolism regulation in artificial cells. **To improve the autonomous**

properties of current artificial cells, an ATP regeneration module can be encapsulated. [...]

2. Figure 1 would benefit from overall improvement in quality. Several abbreviations used in the main text do not align with the presented figure, requiring back-and-forth referencing, thereby reducing the readability. In addition, the separation between cAMP production and the downstream glycogenolytic pathway is not clearly delineated, making it difficult to grasp the conceptual flow at a glance. It is recommended to consider moving the abbreviations to the figure caption and reorganizing the layout to enhance clarity and logical structure.

Thank the reviewer for the comments and valuable suggestions. We followed the reviewer's suggestion and improved the quality of Figure 1. We revised the abbreviations in the main text to align with Figure 1. We used different color arrows to distinguish the signal transduction from ISO to cAMP (orange arrows) and the subsequent glycogenolytic metabolism (blue arrows). The abbreviations were moved to the figure caption as below in page 5.

Fig. 1. Schematic illustration of an artificial cell capable of signal transduction mediated by ADRB2 for the regulation of glycogenolytic metabolism. (a) The signal transduction from extracellular ISO into intracellular cAMP and subsequent regulation of glycogenolytic metabolic pathway inside an artificial cell; ISO: isoproterenol; ADRB2: β_2 -adrenergic receptor; Gs α : Gs subunit α ; ADCY5: adenylyate cyclase V; PKA: Protein kinase A; PhK: Phosphorylase kinase; PYGM: Glycogen phosphorylase; PGM: Phosphoglucomutase; G6PDH: Glucose-6-phosphate dehydrogenase; GDP: Guanosine diphosphate; GTP: Guanosine triphosphate; ATP: Adenosine triphosphate; cAMP:

Cyclic adenosine monophosphate; G-1-P: Glucose 1-phosphate; G-6-P: Glucose 6-phosphate; 6-PGL: 6-Phosphoglucono-lactone; NADP⁺(H): Nicotinamide adenine dinucleotide phosphate. (b) Chemical reaction equations involved in the signal transduction and glycogenolytic metabolism; ADCY5: Adenylate cyclase V; p-PYGM: Phosphorylated glycogen phosphorylase.

3. The presentation of the system's behavior in solution followed by its application in GUV membranes facilitates understanding. However, it also raises the question of why membrane incorporation is essential. Providing insights into the native permeability of the GUV membrane, or any related characterization data, could further support the rationale and strengthen the overall narrative.

Thank the reviewer for the comments and valuable suggestion. The purpose of this work is to build an artificial cell (cell-like structure), which transduce the extracellular signal inside to regulate the internal metabolic pathway. The solution assay was to confirm the viability of signaling pathway and to obtain the optimized condition for GUV membrane reconstitution. In view of artificial cell construction, the membrane incorporation of ADRB2-Gs α and ADCY5 is essential. What is more, the lipid bilayer membrane provides physiological conditions for maintaining the correct conformation of those membrane proteins, and their functions. The lipid bilayer membrane does not allow the penetration of enzymes and ISO. In order to investigate the signal transduction from external to internal of artificial cells, and further regulate the internal metabolic pathway, ADRB2-Gs α and ADCY5 were reconstituted into the GUV membranes. We followed the reviewer's suggestion and carried out experiments on the permeability of GUVs membranes. We added **Supplementary Fig. 8** and corresponding description in page 14.

[...] **No penetration of ISO across the lipid bilayer membrane was detected (Supplementary Fig. 8h).** The coreconstitution of ADRB2-Gs α complexes (Fig. 4b, left image) and ADCY5 (Fig. 4b, middle image) on artificial cells was visualized by fluorescence microscopy. [...]

Supplementary Fig. 8. Representative HPLC chromatograms of ISO at concentrations of 100 μM (a), 200 μM (b), 300 μM (c), 400 μM (d), 600 μM (e), and 800 μM (f). (g) The calibration curve of ISO detected by HPLC. (h) The monitor of concentration of ISO outside GUVs as function of time from 10 to 240 min.

4. Diffusion coefficient of ADRB2 in the GUV appears to be higher than that in the native cell membrane. It would be important to provide additional experiments or, at the very least, a discussion to address the underlying reasons for this difference and how it may influence signal transduction dynamics.

Thank the reviewer for the comment and valuable suggestion. In native cell membranes, physical

constraints and molecular interactions restrict the diffusion of membrane proteins. The diffusion of GPCRs is constrained by direct anchoring and spatial hindrance from the cytoskeleton (e.g., actin filaments, microtubules) and lipid rafts.^{1,2} In contrast, the GUV membrane was composed with pure phospholipids (DOPC/DOPG/CHS), resulting in higher diffusion coefficient of membrane proteins. The faster diffusion of ADRB2-GsaI complexes and ADCY5 enabled the production of cAMP. We added below sentences in page 10-11.

[...] The diffusion coefficient of ADRB2 in the GUV membrane was determined to be 7.81×10^{-9} cm²/s, which was higher than that of ADRB2 on the natural cell membrane (10^{-10} - 10^{-11} cm²/s). Compared with natural cell membranes, artificial cell membranes lack the constraints of the cytoskeletal network³² and lipid rafts³³ because of their simple lipid compositions (DOPC/DOPG/CHS), which explains the faster diffusion coefficient of ADRB2 in GUV membranes. [...]

(1) Alenghat, F. J.; Golan, D. E. Chapter Three - Membrane Protein Dynamics and Functional Implications in Mammalian Cells. In *Current Topics in Membranes*, Bennett, V. Ed.; Vol. 72; Academic Press, 2013; 89-120.

(2) Sevesik, E.; Schütz, G. J. With or without rafts? Alternative views on cell membranes. *BioEssays* 2016, 38 (2), 129-139.

5. It is quite remarkable that both ADRB2-GsaI complexes and ADCY5 are incorporated into artificial cells with such high yield. Is this expected? If so, it might be better to provide some detail on how these are experimentally incorporated into the GUV.

Thank the reviewer for the comment and valuable suggestion. The high co-reconstitution efficiency of 97.4% was obtained by optimizing the reconstitution method including the detergents types and molar ratios of lipids to membrane proteins. Only detergent of LMNG successfully transferred ADRB2-Gsa and ADCY5 onto GUV membranes. The optimal ratio of lipids to membrane proteins was found to be 1000:1 (lipid : protein). We followed reviewer's suggestion and added corresponding description in the section of methods in page 31 as below.

[...] The artificial cells were prepared using gel-assisted hydration method.³⁴ PVA (0.2 g) was dissolved in pure water (10 mL) by stirring at 90 °C for 2 h. The PVA solution was filtered through a 0.22 µm hydrophilic polyethersulfone (PES) membrane to remove insoluble aggregates. PVA film was formed on the bottom of a glass vial by evaporating 100.0 µL PVA solution (0.02 g/mL) at 50 °C for 4 hours. Subsequently, 10.0 µL of phospholipid chloroform solution (0.9 mg/mL DOPC, 0.1 mg/mL DOPG, and 0.05 mg/mL CHS) was evenly spread onto the dried PVA film. The vial was then subjected to vacuum desiccation at 25°C for 2 hours to remove residual organic solvent to form a dry lipid film. Reconstitution buffer containing ADRB2 (20.0 mM HEPES pH 7.4, 100.0 mM NaCl, 0.8 mM ATP, 2.0 mM MgCl₂, 300.0 mM sucrose) was added into the abovementioned vial with a lipid-to-ADRB2 molar ratio ranging from 10²:1 to 10⁶:1. To remove LMNG bound to ADRB2, pretreated Bio-Beads SM-2 were added to the solution at a mass ratio of 1:5 (beads : buffer). The mixture was gently shaken at 30 rpm on a shaker at 4°C for 2 h. The beads were then removed by centrifugation at 200×g for 5 min. The resulting artificial cells were collected and washed three times with the buffer (20.0 mM HEPES pH 7.4, 100.0 mM NaCl, 0.8 mM ATP, 2.0 mM MgCl₂, 300.0 mM glucose) via centrifugation at 1000×g for 5 min at 4°C to remove non-reconstituted

proteins. The similar protocol was applied to fabricate artificial cells reconstituted with ADRB2-Gs α , ADCY5, and ADRB2-Gs α +ADCY5, respectively. [...]

6. In line 267, ...N-terminal “enhanced” yellow fluorescent protein? Is this correct?

Thank the reviewer for the comments. We are sorry about the confusion. The two FRET pair proteins in Epac1-cAMP are called enhanced yellow fluorescent protein and enhanced cyan fluorescent protein. We modified the description as below page 14.

[...] The Epac1-cAMP FRET probes interacted with cAMP to change their conformation to enlarge the distance between enhanced yellow fluorescent protein (EYFP) **linked to N-terminal** and enhanced cyan fluorescent protein (ECFP) **linked to C-terminal**, leading to the simultaneous intensity increase of ECFP and the intensity decrease of EYFP (Supplementary Fig. 11). [...]

7. Why does the phosphorylation percentage of PYGM decrease after 0.8 mM of ATP?

Thank the reviewer for the valuable suggestion. The phosphorylation percentage of PYGM was determined by the concentration of p-PhK, which was determined by the concentration of PhK and ATP. At higher concentration of ATP (>0.8 mM), the conformation of PhK may be influenced by ATP to inhibit its phosphorylation.³⁻⁵ We added below sentences in page 19.

[...] At an ATP concentration of 0.8 mM, the phosphorylation percentage was $90.84 \pm 2.58\%$. **The decline of phosphorylation percentage at higher concentration of ATP may be caused by the conformation change of PhK^{35,36} to prevent its further phosphorylation, consequently decreasing the production of p-PYGM.** Thus, 0.8 mM ATP, 1.0 U of PKA, 1.4 U of PhK, and 1.0 U of PYGM were selected for subsequent glycogenolysis. [...]

(3) Arends, C. J., Wilson, L. H., Estrella, A., Kwon, O. S., Weinstein, D. A., & Lee, Y. M. A Mouse Model of Glycogen Storage Disease Type IX-Beta: A Role for Phkb in Glycogenolysis. *International Journal of Molecular Sciences* 2022, 23(17), 9944.

(4) Bai, Y.; Li, X.; Zhang, D.; Hou, C.; Zheng, X.; Chen, L.; Ren, C. Effects of different ATP contents on phosphorylation level of glycogen phosphorylase and its activity in lamb during incubation at 4 celcius in vitro. *International Journal of Food Science and Technology* 2020, 55 (8), 3000-3007.

(5) Mueller, M. S. Functional impact of glycogen degradation on astrocytic signalling. *Biochemical Society Transactions* 2014, 42, 1311-1315.

8. How does the size of artificial cells influence the experimental outcomes? Additionally, it would be helpful to clarify whether and how the size was controlled or standardized during the experiments.

Thank the reviewer for the comments and the valuable suggestion. We followed the reviewer's suggestion and carried out experiments on the artificial cells size influence on the signal transduction. To evaluate the size effect for the cAMP production, artificial cells with diameter of 3 μm and 10 μm were obtained by extruding the artificial cells through polycarbonate filter membranes with pore sizes of 3 μm or 10 μm . No significant difference of the final cAMP concentration was observed between these two types artificial cells. We added **Supplementary Fig.**

15-17 and corresponding description in page 16. The size control methods were added in page 32.

Page 16

[...] The results from fluorescence spectroscopy indicated the exact same pattern (Fig. 4l) as that from the fluorescence microscopy. No significant difference in the final cAMP concentration was observed between artificial cells with a diameter of 3 μm and those with a diameter of 10 μm (Supplementary Figs. 15-17). [...]

Supplementary Fig. 15. The representative time-series images of artificial cells ($d=3.0 \mu\text{m}$) stimulated by $1.0 \mu\text{M}$ ISO for the intracellular cAMP production at 0, 25, 30 min, with top row images taken by blue channel and bottom row images taken by green channel. The scale bar is $10.0 \mu\text{m}$.

Supplementary Fig. 16. The representative time-series images of artificial cells ($d=10.0 \mu\text{m}$) stimulated by $1.0 \mu\text{M}$ ISO for the intracellular cAMP production at 0, 25, and 30 min, with top row images taken by blue channel and bottom row images taken by green channel. The scale bar is $10.0 \mu\text{m}$.

Supplementary Fig. 17. The cAMP concentration in artificial cells with diameter of 3.0 µm and 10.0 µm at 30 min with the addition of 1.0 µM ISO. The error bars indicate the mean ± standard deviation (SD) (n = 3). nsp > 0.05, *p < 0.05, **p < 0.01, ***p < 0.001 and ****p < 0.0001 by a two-tailed t-test. p < 0.05 was considered statistically significant.

Page 32

[...] To control the size of artificial cells, the artificial cells were further extruded through polycarbonate filter membranes with pore sizes of 3 µm or 10 µm for three times, respectively. The size distribution of the artificial cells was determined by analyzing microscopy images of at least 300 individual GUVs using ImageJ software. [...]

9. Lastly, it would be helpful if the authors could further emphasize the significance of reconstructing the glycogenolytic pathway starting from cAMP. Clarifying both the rationale behind selecting this specific signaling cascade and the evidence supporting the successful reconstitution would strengthen the overall impact and relevance of the study.

Thank the reviewer for the comment and valuable suggestion. In mammalian cells, cAMP functions as a classical second messenger. One of its core physiological roles is the regulation of energy metabolism through the glycogenolysis pathway. When the body is under stress conditions such as exercise or hypoglycemia, adrenaline binds to the ADRB2 on the cell membrane to activate the generation of cAMP. cAMP subsequently activates PKA-PhK-PYGM pathway, leading to the breakdown of glycogen into G-1-P to ultimately provide energy for the cell. In this study, we reconstituted the complete pathway from the extracellular primary messenger (ISO) to the secondary messenger, subsequently to the downstream glycogenolysis. The whole pathway reconstitution in the artificial cells laid the foundation for the construction of autonomy of artificial cells in the future. We also followed the reviewer's suggestion and carried out experiments to confirm the reconstruction of the glycogenolytic pathway in artificial cells. We emphasized the significance reconstructing the glycogenolytic pathway starting from cAMP, added the **Supplementary Fig. 18-19** and corresponding description in page 18 and pages 19- 20 as blow.

Page 18

[...] cAMP is a central secondary messenger and coordinates diverse physiological metabolic pathways, including the glycogenolytic pathway. **Glycogenolysis is a vital metabolic process in organisms that efficiently and rapidly provides energy for life activities.** To regulate the glycogenolytic pathway inside artificial cells through the established ISO signal transduction pathway, a glycogenolytic pathway in solution was constructed. [...]

[...] The glycogenolytic pathway was encapsulated in artificial cells. Artificial cells capable of converting glycogen to G-1-P were constructed with a lumen solution composed of 1.0 μM cAMP, 0.8 mM ATP, 0.5 mM glycogen, 1.0 U/mL PKA, 1.4 U/mL PhK, and 1.0 U/mL PYGM. The reaction was initiated by rapidly increasing the temperature from 4 $^{\circ}\text{C}$ to 37 $^{\circ}\text{C}$. The concentration of G-1-P gradually increased from 0 to 8 minutes and plateaued at a maximum value of $63.28 \pm 3.07 \mu\text{M}$ (Supplementary Fig. 18, purple curve). No G-1-P was produced in the absence of PKA (Supplementary Fig. 18, orange curve) or PhK (Supplementary Fig. 18, blue curve). Artificial cells capable of converting glycogen to 6-PGL were constructed by further encapsulating 0.2 mM NADP^+ , 1.0 U/mL PGM, and 1.0 U/mL G6PDH. In the presence of both PGM and G6PDH, the concentration of NADPH increased within 0–10 minutes to reach a plateau of $129.21 \pm 3.68 \mu\text{M}$ (Supplementary Fig. 19, purple curve). No NADPH was generated in the absence of either PGM (Supplementary Fig. 19, blue curve) or G6PDH (Supplementary Fig. 19, orange curve). These results confirmed the successful reconstitution of the glycogenolysis pathway in artificial cells. [...]

Supplementary Fig. 18. The concentration of produced G-1-P as a function of time inside artificial cells containing 1.0 μM cAMP, 0.8 mM ATP, 0.5 mM glycogen, 1.0 U/mL PKA, 1.4 U/mL PhK, 1.0 U/mL PYGM (purple curve), as well as the control groups without PKA (orange curve) or PhK (blue curve).

Supplementary Fig. 19. The concentration of produced NADPH as a function of time inside artificial cells containing 1.0 μM cAMP, 0.8 mM ATP, 0.5 mM glycogen, 0.2 mM NADP^+ , 1.0 U/mL PKA, 1.4 U/mL PhK, 1.0 U/mL PYGM, 1.0 U/mL PGM, and 1.0 U/mL G6PDH (purple curve), as well as the control groups without G6PDH (orange curve) or PGM (blue curve).

Reviewer #2 (Remarks to the Author):

The manuscript by Liu et al presents the development of artificial cells equipped with a GPCR for transmembrane signaling and with a 5-enzyme intracellular signaling cascade. This study is presented in the context of “bottom up engineering of artificial cells”. As the manuscript states in the opening, artificial cells are designed to understand the working mechanism of mammalian cells. Transmembrane signaling in synthetic cells has indeed been a long standing challenge and as such the subject of this manuscript is indeed very important.

I find the manuscript very interesting but the level of presentation is disappointing. I cannot recommend this manuscript for publication.

1. One critique I have is that the manuscript is written in a “broken” English and in far too many instances, the sentences are grammatically incorrect. One can guess the overall meaning, but an article should not leave room for guessing.

Thank the reviewer for the comment and valuable suggestion. The English has been improved by professional language editing service “Nature publishing group language editing service”.

SPRINGER NATURE
Author Services Editing Certificate

This document certifies that the manuscript

An artificial cell capable of signal transduction mediated by β 2-adrenergic receptor
for the regulation of glycogenolytic metabolism

prepared by the authors

Yanhao Liu, Wan Zhao, Yingming Zhao, Xiangxiang Zhang, Jingjing Zhao, Shubin Li,
Yongshuo Ren, Xiaojun Han

was edited for proper English language, grammar, punctuation, spelling, and overall style
by one or more of the highly qualified English speaking editors at SNAS.

This certificate was issued on **November 10, 2025** and may be verified
on the SNAS website using the verification code **D107-BOE8-F47F-7A4B-C12A**.

Neither the research content nor the authors' intentions were altered in any way during the editing process. Documents receiving this certification should be English-ready for publication; however, the author has the ability to accept or reject our suggestions and changes. To verify the final SNAS edited version, please visit our verification page at secure.authorservices.springernature.com/certificate/verify.
If you have any questions or concerns about this edited document, please contact SNAS at support@es.springernature.com.

2. The second major point of criticism is that the manuscript lacks a clear mission statement, an explanation of exactly what are the challenges that the manuscript addresses?

What has been done in the field, what were the failures that this manuscript overcomes ?

I find that the introduction does not present the state of art, does not explain the challenges, and does not present the prior steps taken to address these challenges. As such, the experimental work is presented as if in a vacuum of knowledge on the subject.

In my experience, the transfer of GPCR from mammalian to artificial cells has been a major challenge. But as presented, the manuscript does not explain, what has been the challenge ? what proteins have been reconstituted within artificial cells ? what methods are there to do this ?

Thank the reviewer for the comment and valuable suggestion. The challenge addressed by this paper is the regulation of intracellular downstream metabolism through GPCR signal transduction in artificial cells. We followed the reviewer's suggestion and thoroughly revised the introduction as below.

[...] Previous attempts at the signal transduction of artificial cells have relied mainly on synthetic receptors, which mimic the working mechanism of natural receptors, including receptor tyrosine kinases (RTKs) and G-protein-coupled receptors (GPCRs).⁹⁻¹¹ RTKs transduce signals through ligand-induced receptor dimerization mechanisms. Cysteine-modified cholesterol¹² and lithocholic acid derivatives¹³ were synthesized as receptors to mimic RTKs, which were dimerized via supramolecular interactions to transduce signals from outside artificial cells to inside them. Anionic ethylenediamine-based receptors were reconstituted into artificial cell membranes, which were dimerized by extracellular cationic ligands via electrostatic interactions to generate intracellular fluorescent signals.¹⁴ Transmembrane DNA strands were also used as RTK analogues anchored on the vesicle membrane¹⁵. Extracellular DNA-induced dimerization triggered intracellular DNAzyme generation, cleaving quenched substrates to produce fluorescence. The conformational changes of GPCRs were mimicked with synthesized azobenzene-modified foldamers containing photoresponsive properties for signal transduction in artificial cells.¹⁶ Extracellular light induced local conformational changes in the photosensitive head group of the foldamer, inducing global conformational chiral changes in the receptors. Cholesterol-modified triplex DNA (TD), as a synthetic receptor, transmits pH signals across lipid bilayers via H⁺-mediated TD conformational transitions.¹⁷ These receptors fail to generate secondary messengers to regulate biological downstream metabolism.

GPCRs constitute the largest family of membrane receptors that mediate responses to a wide variety of hormones, neurotransmitters, and other environmental stimuli.¹⁸⁻²⁰ GPCRs, including the β 2-adrenergic receptor (ADRB2),²¹⁻²³ A_{2A} adenosine receptor (A_{2A}AR),^{24,25} and dopamine receptor²⁶, were reconstituted into phospholipid vesicle membranes to study their conformational changes stimulated by ligands and ligand-binding properties. Upon agonist binding, transmembrane helix VI of A_{2A}AR exhibited rotation and outwards displacement in the membrane of lipid vesicles.²⁵ Dopamine receptor D2 was reconstituted into the membrane of polymer vesicles such that the half maximal effective concentration (EC₅₀) of dopamine was 30 μ M.²⁷ No secondary messengers were produced inside artificial cells. The regulation of intracellular downstream metabolism through GPCR signal transduction triggered by extracellular ligands remains a great challenge in the field of artificial cells. [...]

3. There is a significant mis-match between the ambition of the manuscript to be published in Nature Communications and the data presentation: It is presented as if the GPCR signaling pathway is transplanted into an artificial cell and it works via routine protocols, without much optimization, it just does. So why is this worth Nature Communications if it does not address a challenge and is so easy ?

Thank the reviewer for the comment. First of all, we have to clarify that this work addressed the big challenge in the field of artificial cells after long time hard work. This study took us 3.5 years to complete the experiments, during which a lot of difficulties were solved via systematical

optimization.

The successful reconstitution of signaling pathway was the result of a systematic, multi-stage optimization process with three sequential phases.

Phase 1: Optimization of the signaling pathway in the solution (Fig. 2):

We first investigated the experimental conditions of ISO-to-cAMP signaling pathway involved with ADRB2, G α , and ADCY5 in solution. The optimal catalytic conditions of enzyme ADCY5 were studied to obtain the optimal pH of 8.0, temperature of 37°C, and Mg²⁺ concentration of 2.0 mM. The Michaelis-Menten constants and V_{max} was subsequently obtained to be 144.5 μ M, and 171.4 μ M min⁻¹ for ATP conversion, respectively. The molar ratio of the pre-formed ADRB2-G α complex to ADCY5 was optimized to obtain the ratio (17.4:10.0) with maximum cAMP production.

Phase 2: Optimization of reconstitution methods (Fig. 3):

We tried to reconstitute the membrane proteins using phase transfer method, but failed. We finally succeeded using gel-assisted hydration method to reconstitute ADRB2-G α and ADCY5 after many attempts. The high co-reconstitution efficiency of 97.4% was obtained by optimizing the reconstitution method including the detergent types and molar ratios of lipids to membrane proteins. Only detergent of LMNG successfully transferred ADRB2-G α and ADCY5 onto GUV membranes. The optimal ratio of lipids to membrane proteins was found to be 1000:1 (lipid : protein).

Phase 3: Optimization of the glycogenolytic pathway (Fig. 5):

We investigated the enzyme concentrations of PKA, PhK and PYGM involved glycogenolytic pathway from glycogen to G-1-P using western-blotting method. The optimal concentrations of PKA, PhK and PYGM were determined to be 1.0 U/mL, 1.4 U/mL, and 1.0 U/mL, respectively. The optimal ATP concentration was found to be 0.8 mM.

4. Introduction is not matched to the manuscript. The whole work is about reconstituting proteins. For this reason, I find that the introduction about artificial ion channels is redundant, because (to my knowledge) none of these examples of transmembrane signaling resulted in a biologically – relevant response within the artificial cell.

Subsequent paragraph on RTK and GPCR mimics is not incorrect but misses some critical references in which biologically relevant response has been achieved.

Thank the reviewer for the comment and valuable suggestion. We followed the reviewer's suggestion and deleted the artificial ion channel paragraph. The introduction was thoroughly revised as below.

[...] Previous attempts at the signal transduction of artificial cells have relied mainly on synthetic receptors, which mimic the working mechanism of natural receptors, including receptor tyrosine kinases (RTKs) and G-protein-coupled receptors (GPCRs).⁹⁻¹¹ RTKs transduce signals through ligand-induced receptor dimerization mechanisms. Cysteine-modified cholesterol¹² and lithocholic acid derivatives¹³ were synthesized as receptors to mimic RTKs, which were dimerized via supramolecular interactions to transduce signals from outside artificial cells to inside them. Anionic ethylenediamine-based receptors were reconstituted into artificial cell membranes, which were dimerized by extracellular cationic ligands via electrostatic interactions to generate intracellular

fluorescent signals.¹⁴ Transmembrane DNA strands were also used as RTK analogues anchored on the vesicle membrane¹⁵. Extracellular DNA-induced dimerization triggered intracellular DNAzyme generation, cleaving quenched substrates to produce fluorescence. The conformational changes of GPCRs were mimicked with synthesized azobenzene-modified foldamers containing photoresponsive properties for signal transduction in artificial cells.¹⁶ Extracellular light induced local conformational changes in the photosensitive head group of the foldamer, inducing global conformational chiral changes in the receptors. Cholesterol-modified triplex DNA (TD), as a synthetic receptor, transmits pH signals across lipid bilayers via H⁺-mediated TD conformational transitions.¹⁷ These receptors fail to generate secondary messengers to regulate biological downstream metabolism.

GPCRs constitute the largest family of membrane receptors that mediate responses to a wide variety of hormones, neurotransmitters, and other environmental stimuli.¹⁸⁻²⁰ GPCRs, including the β 2-adrenergic receptor (ADRB2),²¹⁻²³ A_{2A} adenosine receptor (A_{2A}AR),^{24,25} and dopamine receptor²⁶, were reconstituted into phospholipid vesicle membranes to study their conformational changes stimulated by ligands and ligand-binding properties. Upon agonist binding, transmembrane helix VI of A_{2A}AR exhibited rotation and outwards displacement in the membrane of lipid vesicles.²⁵ Dopamine receptor D2 was reconstituted into the membrane of polymer vesicles such that the half maximal effective concentration (EC₅₀) of dopamine was 30 μ M.²⁷ No secondary messengers were produced inside artificial cells. The regulation of intracellular downstream metabolism through GPCR signal transduction triggered by extracellular ligands remains a great challenge in the field of artificial cells. [...]

Results and discussion:

5. The manuscript as presented does not present sufficient level of experimental details to reproduce this work. As such, the process of creating GUVs, the incorporation of transmembrane proteins and that for the intracellular proteins are discussed hardly at all. With all the experience of my lab in this field, we would not be able to walk to the lab and reproduce this work. starting with the formation of GUV.

Thank the reviewer for the comment and valuable suggestion. We followed the reviewer's suggestion and added experimental details in the Methods section as below.

Pages 31-32

Construction of artificial cells capable of signal transduction mediated by ADRB2

The artificial cells were prepared using gel-assisted hydration method.³⁴ PVA (0.2 g) was dissolved in pure water (10 mL) by stirring at 90 °C for 2 h. The PVA solution was filtered through a 0.22 μ m hydrophilic polyethersulfone (PES) membrane to remove insoluble aggregates. PVA film was formed on the bottom of a glass vial by evaporating 100.0 μ L PVA solution (0.02 g/mL) at 50 °C for 4 hours. Subsequently, 10.0 μ L of phospholipid chloroform solution (0.9 mg/mL DOPC, 0.1 mg/mL DOPG, and 0.05 mg/mL CHS) was evenly spread onto the dried PVA film. The vial was then subjected to vacuum desiccation at 25°C for 2 hours to remove residual organic solvent to form a dry lipid film. Reconstitution buffer containing ADRB2 (20.0 mM HEPES pH 7.4, 100.0 mM NaCl, 0.8 mM ATP, 2.0 mM MgCl₂, 300.0 mM sucrose) was added into the abovementioned vial with a lipid-to-ADRB2 molar ratio ranging from 10²:1 to 10⁶:1. To remove LMNG bound to ADRB2, pretreated Bio-Beads SM-2 were added to the solution at a mass ratio of 1:5 (beads :

buffer). The mixture was gently shaken at 30 rpm on a shaker at 4°C for 2 h. The beads were then removed by centrifugation at 200×g for 5 min. The resulting artificial cells were collected and washed three times with the buffer (20.0 mM HEPES pH 7.4, 100.0 mM NaCl, 0.8 mM ATP, 2.0 mM MgCl₂, 300.0 mM glucose) via centrifugation at 1000×g for 5 min at 4°C to remove non-reconstituted proteins. The similar protocol was applied to fabricate artificial cells reconstituted with ADRB2-Gsα, ADCY5, and ADRB2- Gsα+ADCY5, respectively.

To control the size of artificial cells, the artificial cells were further extruded through polycarbonate filter membranes with pore sizes of 3 μm or 10 μm for three times, respectively. The size distribution of the artificial cells was determined by analyzing microscopy images of at least 300 individual GUVs using ImageJ software. The density of artificial cells was directly quantified using an automated cell counter, by adding artificial cell solution (10 μL) to the cell counting chamber.

Pages 32-33

Assay of the percentage of GUVs containing ADRB2-Gsα complex and ADCY5

Flow cytometry was used for determining the percentage of GUVs containing ADRB2-Gsα complex and ADCY5. GUVs without membrane proteins served as blank controls. The 660/20 nm bandpass filter and 525/40 nm bandpass filter were used for Cy5 labelled ADRB2-Gsα detection and FITC labelled ADCY5 detection, respectively. The flow rate was set to 10 μL/min to ensure single GUV detection. The threshold was adjusted based on forward scatter (FSC) and side scatter (SSC) to exclude small lipid aggregates, with only events with FSC-A >10⁴ and SSC-A >10³ to be GUVs. The flow cytometry data acquisition for each sample included 10⁴ GUVs. GUVs without membrane proteins were analyzed to establish the autofluorescence background with a threshold to be 99% of this population. GUVs with membrane proteins exhibiting fluorescence intensity above this threshold were identified as positive samples. The percentage of GUVs containing membrane proteins was calculated using following equation:

$$P = \frac{N_{positive}}{N_{total}} \times 100\%$$

where $N_{positive}$ is the number of GUVs containing membrane proteins, and N_{total} is the number of total GUVs.

Pages 34-35

Assay of cAMP in artificial cells

The artificial cells were reconstituted with ADRB2-Gsα and ADCY5 and with the lumen solution of 20.0 mM HEPES pH 8, 300.0 mM sucrose, 100.0 mM NaCl, 0.5 mM ATP, 2.0 mM MgCl₂. To visualize the generation of cAMP in individual artificial cells in real-time, the Epac1-cAMPs probe (0.7 μM) was encapsulated within the lumen of GUVs. Time-lapse images were acquired. The GUVs were observed using a laser scanning confocal microscope at 37°C with two channels of ECFP and EYFP excited by a 405 nm laser. The quantitative measurement of cAMP concentration was performed using a fluorescence spectroscopy. Following the addition of various concentration of ISO outside of the artificial cells for 30 minutes, 10% Triton X-100 was introduced to disrupt the artificial membranes, with subsequent addition of 5% trichloroacetic acid (12% w/v) to halt reactions. [...]

Pages 35-36

HPLC assay of G-1-P

G-1-P lacks a strong chromophore for sensitive UV detection. Therefore, prior to HPLC analysis, G-1-P underwent an aniline derivatization. The lysed artificial cell solution was filtrated through a 0.22 μm Nylon 66 membrane to remove proteins. Afterwards, a 200 μL aliquot was mixed with 0.2 M aniline hydrochloride solution (100 μL , pH 2.0). The mixture was incubated at 60 $^{\circ}\text{C}$ for 30 min to allow the Schiff base formation between the aldehyde group of ring-opened G-1-P and the amine group of aniline. The reaction was then quenched by adding 1.0 M NaOH (100 μL). To stabilize the derivative, 0.2 M sodium cyanoborohydride (NaBH_3CN) (200 μL) was added at room temperature for 1 h to reduce the Schiff base to a stable secondary amine. The excess aniline was removed by extracting the mixture with 500 μL of dichloromethane (DCM). The upper aqueous layer containing the derivatized G-1-P was collected, followed by filtering through a 0.22 μm Nylon 66 membrane. Subsequently, 5.0 μL aliquots were injected into the HPLC instrument equipped with a reverse-phase C18 column (4.6 \times 250 mm, 5 μm particle size) at 30 $^{\circ}\text{C}$. H_2O -acetonitrile (90:10, v/v) was used as elution with a flow rate of 1 mL/min. UV absorbance was monitored at 280 nm. The G-1-P concentration was obtained using the calibration curve (Supplementary Fig. 21h).

Pages 36-37

LC-MS assay of G-1-P

With the addition of ISO (1 μM) outside of artificial cells for 30 mins, 10% Triton X-100 was added to lyse artificial cells, followed by adding 5% trichloroacetic acid (12% w/v) to terminate reactions. The sample was centrifuged at 4 $^{\circ}\text{C}$ and 12,000 \times g for 15 min to collect the supernatant, which was filtered through a 0.22 μm Nylon 66 membrane before injection. Chromatographic separation was performed on a BEH Amide column (2.1 mm \times 100 mm, 1.7 μm) at 35 $^{\circ}\text{C}$. Mobile phase A consisted of a 5 mM ammonium acetate aqueous solution (pH 4.5), and mobile phase B was acetonitrile. The gradient elution program was set as follows: 0 – 4 min, linear decrease of phase B from 90% to 70%; 4 – 4.1 min, increase of phase B to 90%; and 4.1 – 6 min, re-equilibration at 90% of phase B, with a flow rate of 0.3 mL \cdot min $^{-1}$. Mass spectrometric detection was carried out using an electrospray ionization source in negative ion mode, with the spray voltage of -3.2 kV, the ion source temperature of 330 $^{\circ}\text{C}$, the sheath gas pressure of 35 psi, the auxiliary gas pressure of 8 psi, and the collision gas (argon) pressure of 1.2 mTorr. The precursor ion (theoretical $m/z=259.0220$, $[\text{M}-\text{H}]^{-}$) was acquired in full-scan mode, and characteristic fragment ions were obtained via MS/MS.

Pages 37-38

Assay of NADPH in the artificial cells reconstituted with ADRB2-Gs α and ADCY5 and encapsulated with glycogenolysis pathway

The artificial cells reconstituted with ADRB2-Gs α and ADCY5 and encapsulated with glycogenolysis pathway were constructed with the lumen solution containing 20 mM HEPES pH 8, 300 mM sucrose, 100.0 mM NaCl, 0.8 mM ATP, 2.0 mM MgCl_2 , 0.5 mM glycogen, 0.2 mM NADP^+ , 1.0 U/mL PKA, 1.4 U/mL PhK, 1.0 U/mL PYGM, 1.0 U/mL PGM, 1.0 U/mL G6PDH. The fluorescence microscopy was used for qualitative visualization of NADPH production within artificial cells. At designated time points after the addition of ISO, 10 μL of the artificial cell solution was placed on a glass slide and immediately observed using an inverted fluorescence microscope with blue channel. The quantitative determination of NADPH concentration was performed using

UV-Vis spectroscopy. To terminate the signaling and metabolic reactions, the artificial cell solution was treated with 10% (v/v) Triton X-100 to release intracellular contents, followed immediately by the addition of trichloroacetic acid (TCA) to a final concentration of 5% (w/v) to denature and precipitate proteins. The resulting lysate was centrifuged at $10,000 \times g$ for 5 min at 4 °C, followed by carefully collecting the supernatant and filtering it through a 0.22 μm nylon 66 membrane. The concentration of NADPH in the supernatant was quantified by measuring its absorbance at 340 nm using a UV-vis spectrophotometer. The NADPH concentration was determined by the calibration curve (Supplementary Fig. 23).

6. GPCR is a transmembrane protein, with seven transmembrane domains; Gs-a is a bilayer bound protein. How is it possible that the ADBR2-Gsa-ADCY5 connection is realized in solution?

Thank the reviewer for the comments. The ADBR2-Gsa-ADCY5 connection was already realized in solution by Ugur et al.⁶ They produced cAMP via the connection of ADBR2-Gsa-ADCY5 triggered by ISO. These membrane proteins were stabilized with detergents in solution. The detergents preserved their native structures and functions.⁷ It is a well-established methodological approach to study the membrane proteins.^{8,9}

(6) Oner, S. S.; Kaya, A. I.; Onaran, H. O.; Ozcan, G.; Ugur, O. β 2-Adrenoceptor, Gs and adenylyl cyclase coupling in purified detergent-resistant, low density membrane fractions. *European Journal of Pharmacology* **2010**, *630* (1), 42-52.

(7) Gilbert G. P. Detergents for the stabilization and crystallization of membrane proteins. *Methods* **2007**, *41*(4), 388-397.

(8) Frezza, E.; Amans, T.-M.; Martin, J. Allosteric Inhibition of Adenylyl Cyclase Type 5 by G-Protein: A Molecular Dynamics Study. *Biomolecules* **2020**, *10* (9), 1330.

(9) Qi, C.; Sorrentino, S.; Medalia, O.; Korkhov, V. M. The structure of a membrane adenylyl cyclase bound to an activated stimulatory G protein. *Science* **2019**, *364* (6438), 389-394.

7. The correct GPCR orientation (over 94% yield !!!) is also presented as if it happens without optimization. This paragraph (lines 187-199) is really sketchy and hard if possible at all to follow.

Thank the reviewer for the comments. We developed our membrane protein reconstitution method based on the previous protocol in Kobilka's paper.¹⁰ They obtained the correct orientation of ADRB2 to be about 90%. In our case, the introduction of Cholesterol hemisuccinate (CHS) may facilitate the correct orientation of ADRB2, since it enhanced GPCR stability and promoted correct orientation through interactions with transmembrane helices.^{11,12} We modified the paragraph (line 187-199) as below in page 10.

[...] The correct orientation of ADRB2 in GUV membranes (the extracellular domain facing outwards) is critical for signal transduction. The extracellular domain of ADRB2 contains an ISO-binding domain and other domains modified with N-glycans. To determine the correct orientation rate of ADRB2 in GUV membranes, we treated GUVs with the addition of PNGase F to cleave glycosidic bonds between asparagine residues and the N-acetylglucosamine of ADRB2. Subsequently, the GUVs were lysed with Triton X-100 to measure the molecular weight of ADRB2. The decrease in molecular weight confirmed the correct orientation of ADRB2 in the GUV membranes (Fig. 3c, top row images), with a correct orientation rate of $94.06 \pm 4.24\%$ determined

by measuring the band intensities (Fig. 3c, bottom graph). [...]

(10) Fung, J. J.; Deupi, X.; Pardo, L.; Yao, X. J.; Velez-Ruiz, G. A.; DeVree, B. T.; Sunahara, R. K.; Kobilka, B. K. Ligand regulated oligomerization of b22 adrenoceptors in a model lipid bilayer. *The EMBO Journal* **2009**, *28* (21), 3315-3328

(11) O'Malley, M. A.; Helgeson, M. E.; Wagner, N. J.; Robinson, A. S. The Morphology and Composition of Cholesterol-Rich Micellar Nanostructures Determine Transmembrane Protein (GPCR) Activity. *Biophysical Journal* **2011**, *100* (2), L11-L13.

(12) Thakur, N.; Ray, A. P. P.; Sharp, L.; Jin, B.; Duong, A.; Pour, N. G.; Obeng, S.; Wijesekara, A. V. V.; Gao, Z.-G.; McCurdy, C. R. R.; et al. Anionic phospholipids control mechanisms of GPCR-G protein recognition. *Nature Communications* **2023**, *14* (1),

8. The assay in Figure 3D is rather critical but is presented in details that leave too much out and is impossible to understand.

Thank the reviewer for the comment and valuable suggestion. BODIPY TR GTP γ S was the fluorescence probe to detect G α . In artificial cells, it was in the aggregate state with no fluorescence. Once it bound to G α on the membrane, it emitted fluorescence. To better illustrate the working mechanism of the probes, we have depicted multiple probes within artificial cells. In the left artificial cell, the probes aggregate in a quenched state, while in the right artificial cell, the probes bind with G α and produce fluorescence. We revised Figure. 3d and corresponding description in pages 10-11.

[...] BODIPY TR GTP γ S was employed as a fluorescent probe to validate ADRB2-G α complex reconstitution in the GUV membrane (Fig. 3d). **In the absence of binding to G α , BODIPY TR-GTP γ S aggregated inside the lumen of artificial cells; therefore, they did not emit fluorescence because of the quenching effect (Fig. 3d, left image). Upon ISO stimulation, G α was activated to recruit BODIPY TR GTP γ S, which consequently emitted red fluorescence (Fig. 3d, right image).** [...]

Fig. 3. (d) Schematic diagram of the G α detection mechanism using BODIPY TR GTP γ S.

9. p.15, production of cAMP is measured as density per mL. What sort of system of measurements

is this ?

Thank the reviewer for the comment. We apologize for causing the confusion. The density mentioned here actually refers to the density of artificial cells. We revised the text in pages 14-15 as below.

[...] At an artificial cell density of 10^7 cells·mL⁻¹, the addition of 1.0 μM ISO to the extracellular solution induced cAMP production within the artificial cells, leading to gradual recovery of ECFP fluorescence (Fig. 4c, top row images) and concurrent attenuation of EYFP emission (Fig. 4c, bottom row images). [...]

[...] After 30 min, the cAMP concentration inside artificial cells (10^7 cells·mL⁻¹) exhibited a three-phase positive correlation with the concentration of ISO from 0.1 to 6.0 μM (Fig. 4f, green curve, and Supplementary Fig. 14). [...]

10. if the connection between the agonist and cAMP is possible in solution, what benefit does an artificial cell have ? why bother making an artificial cell ?

Thank the reviewer for the comments and valuable suggestion. The purpose of this work is to build an artificial cell (cell-like structure), which transduce the extracellular signal inside to regulate the internal metabolic pathway. The solution assay was to confirm the viability of signaling pathway and to obtain the optimized condition for GUV membrane reconstitution. In view of artificial cell construction, the membrane incorporation of ADRB2-Gsα and ADCY5 is essential. What is more, the lipid bilayer membrane provides physiological conditions for maintaining the correct conformation of those membrane proteins and their functions. The lipid bilayer membrane does not allow the penetration of enzymes and ISO. In order to investigate the signal transduction from external to internal of artificial cells, and further regulate the internal metabolic pathway, ADRB2-Gsα and ADCY5 were reconstituted into the GUV membranes.

11. the “final” composition of the GUV includes 3 proteins for transmembrane signaling and 5 for downstream signaling, each possibly with its own cofactors, necessary ions, starting materials, and then products formed. Some of these components are membrane permeable and some are not – has this been controlled, quantifies, analyzed ? The experience of my lab is that these multicomponent mixtures are incredibly difficult to control and handle. Others report manuscript-long optimization of these protocols (see Danelon et al, Nature Comm) This manuscript presents it as a routine protocol.

Thank the reviewer for the comment and valuable suggestion. For the enzymes involved in this work (ADCY5, PKA, PhK, PYGM, PGM, G6PDH), all necessary cofactors and coenzymes were listed in the Table as below. All components were added to the lumen solution before artificial cell encapsulation.

Enzyme	Cofactor
ADCY5	Mg ²⁺
PKA	Mg ²⁺ , ATP
PhK	Mg ²⁺ , ATP, Ca ²⁺
PGM	Mg ²⁺
G6PDH	Mg ²⁺ , NADP ⁺

The successful reconstitution of signaling pathway was the result of a systematic, multi-stage optimization process with three sequential phases.

Phase 1: Optimization of the signaling pathway in the solution (Fig. 2):

We first investigated the experimental conditions of ISO-to-cAMP signaling pathway involved with ADRB2, Gs α , and ADCY5 in solution. The optimal catalytic conditions of enzyme ADCY5 were studied to obtain the optimal pH of 8.0, temperature of 37°C, and Mg²⁺ concentration of 2.0 mM. The Michaelis-Menten constants and V_{max} was subsequently obtained to be 144.5 μ M, and 171.4 μ M min⁻¹ for ATP conversion, respectively. The molar ratio of the pre-formed ADRB2-Gs α complex to ADCY5 was optimized to obtain the ratio (17.4:10.0) with maximum cAMP production.

Phase 2: Optimization of reconstitution methods (Fig. 3):

We tried to reconstitute the membrane proteins using phase transfer method, but failed. We finally succeeded using gel-assisted hydration method to reconstitute ADRB2-Gs α and ADCY5 after many attempts. The high co-reconstitution efficiency of 97.4% was obtained by optimizing the reconstitution method including the detergent types and molar ratios of lipids to membrane proteins. Only detergent of LMNG successfully transferred ADRB2-Gs α and ADCY5 onto GUV membranes. The optimal ratio of lipids to membrane proteins was found to be 1000:1 (lipid : protein).

Phase 3: Optimization of the glycogenolytic pathway (Fig. 5):

We investigated the enzyme concentrations of PKA, PhK and PYGM involved glycogenolytic pathway from glycogen to G-1-P using western-blotting method. The optimal concentrations of PKA, PhK and PYGM were determined to be 1.0 U/mL, 1.4 U/mL, and 1.0 U/mL, respectively. The optimal ATP concentration was found to be 0.8 mM.

12. Figure 6 : how was G6P quantified vis MS and HPLC ? there is no details on these experiments.

Thank the reviewer for the comment and valuable suggestion. G-1-P is the direct product of glycogenolysis catalyzed by PYGM. Therefore, G-1-P concentration was quantified to monitor the glycogenolysis. G-6-P was a downstream intermediate product, which was rapidly catalyzed by

G6PDH in the presence of NADP⁺, therefore NADPH was quantified rather than G-6-P. The experimental details of quantifying G-1-P via HPLC and LC-MS were added in pages 35-36 as below.

Pages 35-36:

HPLC assay of G-1-P

G-1-P lacks a strong chromophore for sensitive UV detection. Therefore, prior to HPLC analysis, G-1-P underwent an aniline derivatization. The lysed artificial cell solution was filtrated through a 0.22 µm Nylon 66 membrane to remove proteins. Afterwards, a 200 µL aliquot was mixed with 0.2 M aniline hydrochloride solution (100 µL, pH 2.0). The mixture was incubated at 60 °C for 30 min to allow the Schiff base formation between the aldehyde group of ring-opened G-1-P and the amine group of aniline. The reaction was then quenched by adding 1.0 M NaOH (100 µL). To stabilize the derivative, 0.2 M sodium cyanoborohydride (NaBH₃CN) (200 µL) was added at room temperature for 1 h to reduce the Schiff base to a stable secondary amine. The excess aniline was removed by extracting the mixture with 500 µL of dichloromethane (DCM). The upper aqueous layer containing the derivatized G-1-P was collected, followed by filtering through a 0.22 µm Nylon 66 membrane. Subsequently, 5.0 µL aliquots were injected into the HPLC instrument equipped with a reverse-phase C18 column (4.6 × 250 mm, 5 µm particle size) at 30°C. H₂O–acetonitrile (90:10, v/v) was used as elution with a flow rate of 1 mL/min. UV absorbance was monitored at 280 nm. The G-1-P concentration was obtained using the calibration curve (Supplementary Fig. 21h).

Page 36-37

LC-MS assay of G-1-P

With the addition of ISO (1 µM) outside of artificial cells for 30 mins, 10% Triton X-100 was added to lyse artificial cells, followed by adding 5% trichloroacetic acid (12% w/v) to terminate reactions. The sample was centrifuged at 4°C and 12,000 × g for 15 min to collect the supernatant, which was filtered through a 0.22 µm Nylon 66 membrane before injection. Chromatographic separation was performed on a BEH Amide column (2.1 mm × 100 mm, 1.7 µm) at 35°C. Mobile phase A consisted of a 5 mM ammonium acetate aqueous solution (pH 4.5), and mobile phase B was acetonitrile. The gradient elution program was set as follows: 0 – 4 min, linear decrease of phase B from 90% to 70%; 4 – 4.1 min, increase of phase B to 90%; and 4.1 – 6 min, re-equilibration at 90% of phase B, with a flow rate of 0.3 mL·min⁻¹. Mass spectrometric detection was carried out using an electrospray ionization source in negative ion mode, with the spray voltage of –3.2 kV, the ion source temperature of 330°C, the sheath gas pressure of 35 psi, the auxiliary gas pressure of 8 psi, and the collision gas (argon) pressure of 1.2 mTorr. The precursor ion (theoretical m/z=259.0220, [M–H]⁻) was acquired in full-scan mode, and characteristic fragment ions were obtained via MS/MS.

13. How were the artificial cells characterized for size and for concentration?

Thank the reviewer for the comment and valuable suggestion. The size of artificial cells was measured using ImageJ software by analyzing at least 300 GUVs in fluorescence microscopy images. The concentration of artificial cells was determined using a cell counter. We added corresponding experimental details in page 32 as below.

[...] The size distribution of the artificial cells was determined by analyzing microscopy images of at least 300 individual GUVs using ImageJ software. The density of artificial cells was directly

quantified using an automated cell counter, by adding artificial cell solution (10 μ L) to the cell counting chamber. [...]

14. Production of NADPH was monitored: as artificial cells, or with lysis ? again, little if any experimental details are provided.

Thank the reviewer for the comment and valuable suggestion. Quantitative measurements of NADPH were performed after lysing artificial cells. We followed reviewer's suggestion and added corresponding experimental details in pages 37-38 as below.

[...] The fluorescence microscopy was used for qualitative visualization of NADPH production within artificial cells. At designated time points after the addition of ISO, 10 μ L of the artificial cell solution was placed on a glass slide and immediately observed using an inverted fluorescence microscope with blue channel. The quantitative determination of NADPH concentration was performed using UV-Vis spectroscopy. To terminate the signaling and metabolic reactions, the artificial cell solution was treated with 10% (v/v) Triton X-100 to release intracellular contents, followed immediately by the addition of trichloroacetic acid (TCA) to a final concentration of 5% (w/v) to denature and precipitate proteins. The resulting lysate was centrifuged at $10,000 \times g$ for 5 min at 4 $^{\circ}$ C, followed by carefully collecting the supernatant and filtering it through a 0.22 μ m nylon 66 membrane. The concentration of NADPH in the supernatant was quantified by measuring its absorbance at 340 nm using a UV-vis spectrophotometer. The NADPH concentration was determined by the calibration curve (Supplementary Fig. 23). [...]

15. How did you go from fluorescence to concentration within the artificial cell ? this is far from trivial.

Thank the reviewer for the comment and valuable suggestion. All the concentrations of cAMP and NADPH were measured after lysing the artificial cells. The fluorescence microscopy images were used to visualize the production of cAMP and NADPH. We added the experimental details for concentration measurements of cAMP and NADPH in pages 34 and pages 37-38 respectively as below.

Page 34

Assay of cAMP in artificial cells

The artificial cells were reconstituted with ADRB2-Gs α and ADCY5 and with the lumen solution of 20.0 mM HEPES pH 8, 300.0 mM sucrose, 100.0 mM NaCl, 0.5 mM ATP, 2.0 mM MgCl₂. To visualize the generation of cAMP in individual artificial cells in real-time, the Epac1-cAMPs probe (0.7 μ M) was encapsulated within the lumen of GUVs. Time-lapse images were acquired. The GUVs were observed using a laser scanning confocal microscope at 37 $^{\circ}$ C with two channels of ECFP and EYFP excited by a 405 nm laser. The quantitative measurement of cAMP concentration was performed using a fluorescence spectroscopy. Following the addition of various concentration of ISO outside of the artificial cells for 30 minutes, 10% Triton X-100 was introduced to lyse the artificial cells, with subsequent addition of 5% trichloroacetic acid (12% w/v) to halt reactions. [...]

Pages 37-38

Assay of NADPH in the artificial cells reconstituted with ADRB2-Gs α and ADCY5 and encapsulated with glycogenolysis pathway

The artificial cells reconstituted with ADRB2-Gs α and ADCY5 and encapsulated with glycogenolysis pathway were constructed with the lumen solution containing 20 mM HEPES pH 8, 300 mM sucrose, 100.0 mM NaCl, 0.8 mM ATP, 2.0 mM MgCl₂, 0.5 mM glycogen, 0.2 mM NADP⁺, 1.0 U/mL PKA, 1.4 U/mL PhK, 1.0 U/mL PYGM, 1.0 U/mL PGM, 1.0 U/mL G6PDH. The fluorescence microscopy was used for qualitative visualization of NADPH production within artificial cells. At designated time points after the addition of ISO, 10 μ L of the artificial cell solution was placed on a glass slide and immediately observed using an inverted fluorescence microscope with blue channel. The quantitative determination of NADPH concentration was performed using UV-Vis spectroscopy. To terminate the signaling and metabolic reactions, the artificial cell solution was treated with 10% (v/v) Triton X-100 to release intracellular contents, followed immediately by the addition of trichloroacetic acid (TCA) to a final concentration of 5% (w/v) to denature and precipitate proteins. The resulting lysate was centrifuged at 10,000 \times g for 5 min at 4 $^{\circ}$ C, followed by carefully collecting the supernatant and filtering it through a 0.22 μ m nylon 66 membrane. The concentration of NADPH in the supernatant was quantified by measuring its absorbance at 340 nm using a UV-vis spectrophotometer. The NADPH concentration was determined by the calibration curve (Supplementary Fig. 23).

16. The produced NADPH is reported in hundred micromolar concentration. This is an incredibly high concentration. I would like to see explicitly how was this number (and all the reported numbers) arrived at, because this concentration is remarkably high.

Thank the reviewer for the comment and valuable suggestion. The concentration of intracellular NADPH was 122 μ M by converting NADP⁺ of 0.2 mM catalyzed by G6PDH. The high concentration of substrate (NADP⁺) explained the high concentration of NADPH (122 μ M).

17. It is admirable that the GUV were analyzed via flow cytometry (fig 3h); these data illustrate that fluorescence can be different by as much as 100 fold. I do not understand how was the data in Figure 3i derived, and how are the error bars to small ?

Thank the reviewer for the comment and valuable suggestion. The data in Figure 3i was calculated using the below equation, where N_{positive} and N_{total} were obtained from the flow cytometry histograms shown in Figure 3h using a standardized gating analysis procedure. The small error bars indicate that the developed protocol of constructing artificial cells with membrane proteins was reliable, because they were systematically optimized. The detailed method was added in pages 32-33 as below.

Assay of the percentage of GUVs containing ADRB2-Gs α complex and ADCY5

Flow cytometry was used for determining the percentage of GUVs containing ADRB2-Gs α complex and ADCY5. GUVs without membrane proteins served as blank controls. The 660/20 nm bandpass filter and 525/40 nm bandpass filter were used for Cy5 labelled ADRB2-Gs α detection and FITC labelled ADCY5 detection, respectively. The flow rate was set to 10 μ L/min to ensure single GUV detection. The threshold was adjusted based on forward scatter (FSC) and side scatter (SSC) to exclude small lipid aggregates, with only events with FSC-A $>10^4$ and SSC-A $>10^3$ to be GUVs. The flow cytometry data acquisition for each sample included 10^4 GUVs. GUVs without

membrane proteins were analyzed to establish the autofluorescence background with a threshold to be 99% of this population. GUVs with membrane proteins exhibiting fluorescence intensity above this threshold were identified as positive samples. The percentage of GUVs containing membrane proteins was calculated using following equation:

$$P = \frac{N_{positive}}{N_{total}} \times 100\%$$

where $N_{positive}$ is the number of GUVs containing membrane proteins, and N_{total} is the number of total GUVs.

18. It would be most appropriate to show microscopy images that with more than one GUV in the field of view;

Thank the reviewer for the comment and valuable suggestion. We provided microscope images of two GUVs with ADRB2-Gs α complex and ADCY5 to replace previous Fig. 4b in page 17 as below.

Fig. 4. (b) Fluorescence images of GUVs containing Cy5 -ADRB2- Gs α complexes and FITC-ADCY5 taken by red channel (left image), green channel (middle image), and their merged image (right image). The scale bar is 10.0 μ m.

19. The natural potency of agonists is typically nanomolar. The fact that all the GPCR activation in this work is conducted using micromolar ISO indicates that the GPCR is not in its native conformation, or is otherwise non-functional.

Thank the reviewer for the comment. The ADRB2 in the artificial cells was confirmed to keep its functions due to following experimental results. The ISO dose-dependent cAMP production was observed (Figures 4e–f) with a maximum cAMP production of 57 μ M. This signal transduction pathway was inhibited by specific antagonists (alprenolol) and inverse agonists (carazolol) (Figures 4h–l). The abovementioned data confirmed that ADRB2 remained its functions.

The lowest ISO concentration to stimulate ADRB2 to produce cAMP was 400 nm in this work. This value is higher than that for native cells, probably due to the far simplicity of artificial cells than native cells.

20. The manuscript critically misses on the explanation of the motivation of this work. Reconstituting the natural components does not reveal any new information about this reaction network. The proposed synthetic cell is simpler but what information does it reveal or make possible to arrive at ?

Thank the reviewer for the comments. The motivation of this work is to build an artificial cell (cell-like structure), which transduce the extracellular signal inside to regulate the internal metabolic

pathway. It is a proof-of-concept work to demonstrate that the GPCR signal transduction is able to be reconstituted in the artificial cells, and further regulation the intracellular downstream metabolism.

In the field of artificial cells, the artificial cells in this work are rather sophisticated, which lay foundation towards autonomous artificial cells. We added below sentences in page 24.

[...] These findings demonstrate the successful reconstitution of the GPCR-mediated signalling pathway for downstream metabolism regulation in artificial cells. **To improve the autonomous properties of current artificial cells, an ATP regeneration module can be encapsulated. The proposed artificial cells lay the foundation for the development of autonomous artificial cells.** [...]

21. Finally, last but not least :

This journal pays close attention to statistical evaluation of data, the reproducibility of work. The data presented throughout the manuscript have remarkably small error bars. Remarkably small.

I would like the authors to clearly state if experiments presented in this work have been reproduced in INDEPENDENT experiments ? as disclosed, does n=3 refer to technical replicates or independent experiments ? if this has not been done in truly independent experiments, it has to be done so.

On the same subject : to my knowledge, t-test is not an appropriate statistical method here and all the data have to be analyzed via ANOVA.

Thank the reviewer for the comment and valuable suggestion. All the experiments in this work were carried out independently at least for three replicates. n=3 refers to 3 independent experiments. The small error bars resulted from the experiment optimization and careful control of experimental conditions.

Student's t-test is suitable for the data of two groups, while one-way ANOVA is suitable for the data with 3 or more groups. We followed the reviewer's suggestion and re-analyzed data with 3 or more groups via ANOVA. The conclusions remained the same. We modified the figure caption of Fig. 2 in page 9, Fig. 3 in page 14, Fig. 5 in page 21 and Fig.6 in page 24, and statistical analysis in Methods section in page 38 as below.

Page 9

[...] The error bars indicate the mean \pm standard deviation (SD) (n = 3). nsp > 0.05, *p < 0.05, **p < 0.01, ***p < 0.001 and ****p < 0.0001 by **one-way ANOVA**. P < 0.05 was considered statistically significant. [...]

Page 14

[...] The error bars indicate the mean \pm standard deviation (SD) (n = 3). nsp > 0.05, *p < 0.05, **p < 0.01, ***p < 0.001 and ****p < 0.0001 by **one-way ANOVA**. P < 0.05 was considered statistically significant. [...]

Page 21

[...] The error bars indicate the mean \pm standard deviation (SD) (n = 3). nsp > 0.05, *p < 0.05, **p < 0.01, ***p < 0.001 and ****p < 0.0001 by **one-way ANOVA**. P < 0.05 was considered statistically significant. [...]

Page 24

[...] The error bars indicate the mean \pm standard deviation (SD) (n = 3). nsp > 0.05, *p < 0.05, **p < 0.01, ***p < 0.001 and ****p < 0.0001 by **one-way ANOVA**. P < 0.05 was considered statistically significant. [...]

Page 38

[...] The significance of differences was analyzed using a two-tailed unpaired Student's t-test for the data of two groups **and one-way ANOVA for the data with 3 or more groups**. [...]

Reviewer #3 (Remarks to the Author):

We really appreciated the review's comments and valuable suggestion, which greatly improved the quality of our manuscript.

For the sake of clarity, the comments of the reviewer have been collated in black, and our response to each comment appears in blue.

Reviewer #1 (Remarks to the Author):

The authors have addressed all the comments and issues that I have raised and I am satisfied with these revisions. Therefore, I recommend publication without further revisions.

We sincerely appreciate Reviewer's recognition and positive feedback on our revision work.

Reviewer #4 (Remarks to the Author):

I have carefully examined the authors' point-by-point responses to Reviewer #2 as well as the corresponding revisions in the manuscript. In my evaluation, the authors have made substantial and appropriate efforts to address the previously raised concerns. They have provided detailed experimental procedures for GUV construction, protein incorporation, and downstream metabolic assays; clarified methodological ambiguities; and added missing analytical details. The quality of the figures has also been improved, and additional experiments have been performed to support key conclusions, particularly regarding signal transduction efficiency and downstream metabolic outputs.

Overall, the technical issues highlighted by Reviewer #2 have been satisfactorily resolved. I believe that the revised manuscript presents a coherent and technically sound study, and in my opinion, it is suitable for publication in Nature Communications in its current form.

We sincerely appreciate Reviewer's recognition and positive feedback on our revision work.